# BRED: A Comprehensive Benchmark for the Robust Evaluation of LLM-Generated Text Detection in Realistic Scenarios

## Abstract

The rapid advancement of large language models (LLMs) has created a pressing need for robust detectors capable of distinguishing between machine-generated and human-written texts. However, existing benchmarks often lack the comprehensive scope needed for rigorous testing. We introduce BRED, a new benchmark that offers four key contributions: 1) extensive coverage of diverse domains and compositional operations, 2) in-depth analysis of LLM pipeline and compositional operations, 3) evaluation across different LLM variants and groups, and 4) in-depth exploration of supervised detectors. Through extensive evaluation of baseline detectors, we have three key findings: 1) supervised LM-based detectors are the most robust against diverse generation strategies, 2) text generated by larger models does not exhibit significant resistance to detection; and 3) current detection methods struggle significantly with texts that have undergone secondary operations from both LLM and operations. BRED provides a standardized platform for assessing detector robustness and offers practical insights for advancing AI-generated text detection. Code and data are available at https://anonymous.4open.science/r/BRED-65D3.

## 1 Introduction

The emergence of large language models (LLMs), such as ChatGPT and DeepSeek, has driven significant progress in text generation. However, their misuse in domains such as academic writing and social media raises concerns, including academic misconduct (Stokel-Walker, 2022) and the dissemination of misinformation (Weidinger et al., 2021; Sison et al., 2024). Consequently, detecting LLM-generated text has become an increasingly critical research topic in recent years.

The rapid advancement of LLMs and their detection methods calls for more comprehensive evaluation frameworks that reflect real-world application scenarios. Currently, several benchmark datasets have been established(He et al., 2024; Guo et al., 2023; Brown et al., 2020; Tao et al., 2024; Wu et al., 2024; Dugan et al., 2024; Zhang et al., 2024; Wang et al., 2023; 2024; Saha & Feizi, 2025). Among these, HC3 (Guo et al., 2023) represents one of the earliest and most widely-used benchmarks, featuring human-written and GPT-3-generated texts in Chinese and English collected primarily from question-answering tasks and encyclopedic sources including Wikipedia and BaiduBaike. CUDRT (Tao et al., 2024) introduced an innovative operational framework encompassing five key text manipulation types: Create, Update, Delete, Rewrite, and Translate, with content in both Chinese and English.

Existing benchmarks have four key limitations that affect their practical utility: 1) They often fail to balance domain and operation coverage, with examples like RAID(Dugan et al., 2024) covering 8 domains but focusing on LLM-created texts, and CUDRT lacking diverse domain data despite offering multiple operations. 2) Real-world usage often involves combinations of operations or LLMs (e.g., generate-then-polish or feeding the output of one LLM into another) rather than single operation or LLM. 3) Benchmarks primarily focus on detecting text from different LLM series, neglecting the quality and detection challenges of texts produced by variants within the same model series. 4) Most benchmarks prioritize zero-shot detection while providing limited analysis of supervised methods.

| Benchmark | Domains | Ops | Supervised | ML | CD | CM | CO | Size | Instruct | Reasoning | LLM-Co | Op-Co |
|-----------|---------|-----|------------|-----|-----|-----|-----|------|----------|-----------|--------|-------|
| MGTBench | 3 | 1 | 12 | ✓ | ✓ | ✓ | × | × | × | × | × | × |
| DetectRL | 4 | 1 | 4 | ✓ | ✓ | ✓ | × | × | × | × | × | × |
| RAID | 8 | 1 | 4 | × | ✓ | ✓ | × | × | × | × | × | × |
| MixSet | 6 | 5 | 6 | × | × | ✓ | ✓ | × | × | × | × | × |
| M4 | 5 | 1 | 5 | ✓ | ✓ | ✓ | × | × | × | × | × | × |
| APT-Eval | 6 | 1 | 4 | × | ✓ | ✓ | × | × | × | × | × | × |
| CUDRT | 4 | 7 | 3 | × | × | ✓ | ✓ | × | × | × | × | × |
| **BRED** | **7** | **7** | **18** | ✓ | ✓ | ✓ | ✓ | ✓ | ✓ | ✓ | ✓ | ✓ |

Table 1: A comparison of data coverage and evaluation tasks. Our provided dataset is the only one that contains diverse combinations of LLM/operation and analysis of LLM groups across recent generative models.

To address these limitations, we introduce **BRED**, a comprehensive benchmark for LLM-generated text detection that spans 7 domains and 7 operations. Our dataset is built with texts produced by 10 LLMs from 4 major families, allowing us to examine both cross-family differences and variations within the same family. To increase practical relevance, we further include diverse combinations of operations/LLMs. We then evaluate a wide range of detection methods, including zero-shot and supervised settings as well as both open-access and commercial detectors, through carefully designed experiments.

In summary, the contributions of **BRED** are as follows:

- A comprehensive and extensible evaluation framework that simultaneously encompasses distinct domains, operations and LLM generators.
- While maintaining the evaluation of zero-shot models, BRED further categorize and investigate the performance of supervised methods, including cutting-edged methods.
- BRED compares performance variations within LLM families by examining three tasks: 1) small vs. large, 2) instruct vs. base, and 3) reasoning vs. non-reasoning.
- BRED defines 17 operational and 12 LLM combinations to systematically investigate how these combinations impact the detectability of LLM-generated text.
- Building on these tasks and evaluations, we derive novel insights that contribute valuable findings to LLM-generated text detection.

## 2 RELATED WORK

In this paper, we mainly focus on black-box detectors. These detectors can be divided into two categories: zero-shot and supervised methods.

**Zero-shot methods**(Yang et al., 2023; Mitchell et al., 2023; Bao et al., 2023; Gehrmann et al., 2019; Beresneva, 2016b) rely on LLMs to extract text features, typically regarded as semantic features, to enable a more comprehensive representation of text information. The principle of these methods involves using pretrained LLMs that have not been fine-tuned on training data to extract syntactic and statistical features of the text, indicating that zero-shot methods can be applied without reliance on training data. The extracted feature vectors are then used to calculate the statistic, which is used to classify the given text. In recent years, zero-shot detection research has developed rapidly. A typical work is DetectGPT(Mitchell et al., 2023), which relies on the log-probability curvature. Building on this, Fast-DetectGPT(Bao et al., 2023) was recently proposed. It adopts the overall model architecture of DetectGPT while optimizing the sampling process and the statistical metric, which improved the detection accuracy and efficiency of DetectGPT.

**Supervised methods**(Liu et al., 2019; Guo et al., 2023; Verma et al., 2023; Chen et al., 2023; Hu et al., 2023b) are highly dependent on training data, which can mainly divded into three categories: 1) **Linguistic-based detectors** models (Guo et al., 2023; Beresneva, 2016a; Verma et al., 2023) extract statistical features from the given text and use them on training. Recently, Ghostbuster(Verma et al., 2023) was proposed, which is based on structured search and linear classification. 2) **Direct LM-based detectors**(He et al., 2021; Yang et al., 2019; Liu et al., 2019; Sanh et al., 2019; Conneau et al., 2019), use pre-trained language models to extract text embeddings for classification. Earlier

work fine-tuned RoBERTa(Liu et al., 2019) for this task, while recent studies show that LLMs like Llama and Mistral also perform well as binary classifiers. 3) **Modified LM-based Detectors**(Yu et al., 2024; Chen et al., 2025; Fu et al., 2025; Tian et al., 2023), utilize augumented pre-trained language models in the detection progress. Among them, the latest ImBDChen et al. (2025) adjusts the logits output of LLMs using Style Preference Optimization(SPO).

## 3 BRED

As shown in Figure 1, BRED uses 7 operations across 7 domain subsets with 4 different LLM families. Further descriptions can be found in Appendix B.

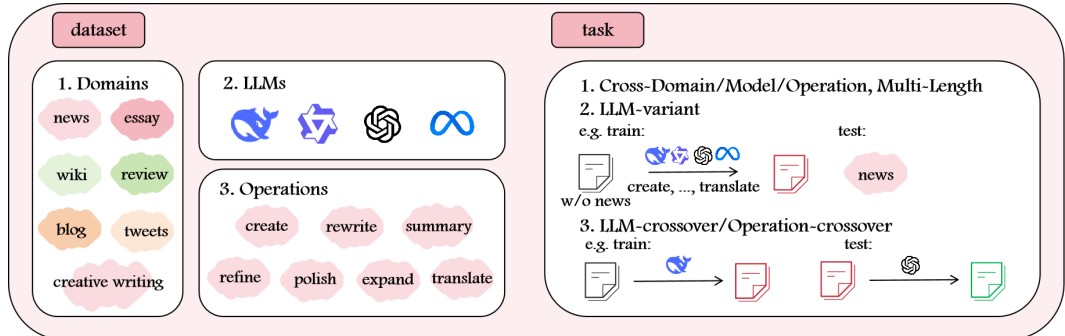

Figure 1: The overview of BRED Benchmark.

### 3.1 DATASET

#### 3.1.1 HUMAN-WRITTEN TEXT COLLECTION

To better reflect real-world scenarios, we categorize target data based on its formality. *Formal texts* include news articles (xsum)(Narayan et al., 2018), Wikipedia entries from SQuAD (squad)(Rajpurkar et al., 2016; 2018), and academic abstracts from PubMedQA (pubmed)(Jin et al., 2019). *Informal texts* include creative writing from WritingPrompts (wp)(Fan et al., 2018), blog articles (blog)(Schler et al., 2006) and short Twitter posts (tweets)(Go et al., 2009). Additionally, to account for the increasing use of LLMs in peer review, we include review comments from OpenReview(review) [5], restricted to those published before 2019.

#### 3.1.2 LLM-BASED TEXT OPERATIONS

Based on previous studies and real-world application scenarios, we employ 7 operations to each domain of human texts to generate LLM-generated texts. These operations are grouped into three main categories: 1) *Creative Reformation*, includes the operations 'create', 'rewrite', and 'summary', which involve generating text from scratch or rephrasing existing content. 2) *Linguistic Enhancement*, consists of 'polish', 'expand', and 'refine', all of which aim to improve or modify the style, clarity, or length of the original text, 3) *Cross-lingual Transfer*, focuses on translating content from one language to another using LLMs. Notably, since our dataset is composed of English texts, we apply the 'translate' operation in two stages. First, each human-written text is translated from English to Chinese, and then translated back into English.

#### 3.1.3 LLMS FOR TEXT GENERATION

For the collected human-written texts, we utilized 4 representative and high-performance LLM families including 10 LLMs to generate corresponding LLM-generated texts for each operation. The models selected include DeepSeek-chat, DeepSeek-reasoner(Liu et al., 2024), gpt4o-mini, gpt4o(OpenAI et al., 2024), gpt5, Qwen-plus, Qwen2-7b-instruct, Qwen3-8b , Llama3.1-8b-v2, and Llama3.1-8b-instruct-v2(Grattafiori et al., 2024). To ensure fairness in comparative evaluations, we used identical corresponding human-written texts for models within the same series or family.

To strike a balance between stability and creativity, while ensuring the generated texts closely mimic human writing in style and coherence, we set the temperature parameter to 0.7 during the generation process. Example prompts used in text generation are illustrated in Appendix B.4.

## 3.2 EVALUATION TASKS

### 3.2.1 OUT-OF-DISTRIBUTION EXPERIMENTS: CROSS-DOMAIN (CD), CROSS-MODEL (CM), CROSS-OPERATION (CO) AND MULTI-LENGTH (ML)

For each OOD experiment, the dataset is divided into subsets based on domain, model, and operation. One subset is used for testing, while the training set is constructed from the remaining subsets, with 4,000 texts selected through stratified sampling. In the Multi-Length experiment, texts are truncated into segments of 10, 50, 100, 200, and 500 words. Details on task configurations and train/test splits are provided in Appendix B.5.

### 3.2.2 LLM VARIANT ANALYSIS (LLM-VA)

The LLM-Variant (LLM-Va) analysis focuses on how inherent model characteristics affect text detectability. This evaluation comprises three comparative subtasks: small vs. large models, instruction-tuned vs. base models, and reasoning vs. non-reasoning models. For each subtask, we used newly incorporated LLMs, with experimental settings similar to the OOD tasks where the dataset was divided into four subsets based on LLM families. Further details on the setup are available in Table B.3.

### 3.2.3 GENERATION COMBINATION ANALYSIS (LLM-CO & OP-CO)

To simulate more realistic and complex usage scenarios, we designed two experiments to analyze the impact of combining generation steps. Detailed data splits are also provided in Table B.4.

**LLM-Crossover (LLM-Co)** This experiment models the scenario where a text generated by one LLM is subsequently reprocessed by another. We constructed 12 distinct model combinations for this task. In each case, the same operation was used in both the initial generation and the reprocessing stage to ensure a controlled comparison. For this experiment, we adopted an in-domain evaluation strategy, dividing the dataset into four subsets based on the LLM used for the initial generation.

**Operation-Crossover (Op-Co)** Similarly, this experiment investigates how applying a sequence of different operations affects detectability. We defined 17 distinct operational combinations where the same LLM was used for both generation steps to isolate the effect of the operations themselves. Following the same in-domain evaluation approach, the dataset was partitioned accordingly.

## 4 DETECTORS

**Zero-shot detectors** We employed three logits-based methods: log-likelihood(Ippolito et al., 2019), rank(Gehrmann et al., 2019), log-rank(Solaiman et al., 2019a), and entropy(Gehrmann et al., 2019). To cover all evaluation aspects, we selected two state-of-the-art zero-shot detectors, DetectGPT(Mitchell et al., 2023) and Fast-DetectGPT(Bao et al., 2023), assessing their performance across diverse datasets.

**Supervised detectors** We selected several detectors from three main categories. Linguistic-based detection models, including GLTR(Guo et al., 2023), perplexity(Beresneva, 2016a), log-likelihood, rank, log-rank, entropy-based classifier, and Ghostbuster(Verma et al., 2023). Direct embedding-based detectors, including DistilBERT, RoBERTa-large, XLM-RoBERTa-large, DeBERTa-v3-large, Llama3.1-8B-v2, Mistral-7B, and XLNet. Modified embedding-based Detectors, including DPIC(Yu et al., 2024), ImBD(Chen et al., 2025), MPU and DetectAnyLLM.

**Open-access detectors** We selected five open-access detectors, using their pretrained checkpoints to evaluate their detection performance, including OpenAI Roberta-based detector(Solaiman et al.,

2019b), RoBERTa-HC3(Guo et al., 2023), RADAR(Hu et al., 2023a), ArguGPT(Liu et al., 2023), and MPU(Tian et al., 2023).

**Commercial detectors**  Since users in real-world scenarios more frequently encounter commercial detection tools, we also incorporated three widely-used commercial detectors: GPTZero[1], ZeroGPT[2], and Winstion AI[3].

## 5 RESULTS AND DISCUSSIONS

In this section, we present our main experimental results, structured around three evaluation tasks: 1) out-of-distribution analysis, 2) performance variations across different LLM groupings, and 3) the impact of LLM/operation crossover experiments. Based on these findings, we derive several practical and novel insights for detector training and selection. To account for potential data leakage and testing costs, the performance of open-access and commercial detectors is reported in separate subsections.

### 5.1 OUT-OF-DISTRIBUTION EXPERIMENTS: CROSS-DOMAIN, CROSS-MODEL, CROSS-OPERATION

Table 2a, 2b and 2c present the results of out-of-distribution (OOD) experiments. The findings show that: 1) Most detectors perform better on informal texts, such as blog posts and tweets, with average AUC scores of 74.44% and 74.53%, respectively. In contrast, formal texts like news articles, academic abstracts, and Wikipedia content are more difficult to detect. 2) Detectors perform worst on Llama-generated text, achieving an average AUC of only 71.09%, significantly lower than other model sources. 3) Detectors perform best on texts generated using the operation 'create'. In contrast, the performance is generally poor on texts from the 'translate', 'refine', and 'summary'. Among these, *translated LLM-generated texts are notably harder to detect*, with an average AUC of 69.60%, making cross-lingual transfer the most challenging. This difficulty may stem from the fact that translated texts preserve the original meaning while undergoing only superficial changes, complicating feature extraction.

As shown in Figure 2 (a), to further discover the influence of text length under ood setting, we introduce multi-length experiment, and draw the following conclusions: 1) The line chart results reveal a clear positive correlation between text length and detection performance (AUC) for most models. 2) However, zero-shot methods like log-likelihood and entropy, along with a few supervised methods such as MPU, demonstrate an opposite trend, with performance deteriorating as text length increases.

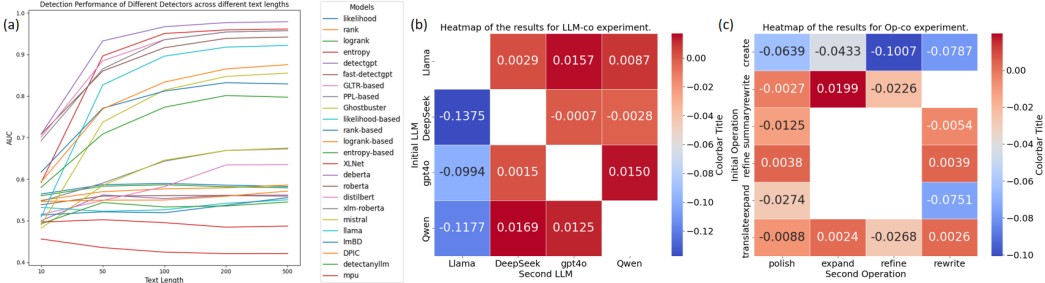

Figure 2: (a) Detection performance of different detectors across different text length (b) The change in detection performance before and after regeneration in LLM-Co experiment. (c) The change in detection performance before and after regeneration in Op-Co experiment.

Table 2: The detection performance of various detectors in Cross-Domain (CD) experiment, Cross-Model(CM), and Cross-Operation(CO) experiments. Detailed results of CM and CO are shown in Appendix C.1.

(a) Cross-Domain (CD)

| AUC | Formal | | | | Informal | | | | | Overall Avg. |
|---|---|---|---|---|---|---|---|---|---|---|
| | xsum | pubmed | squad | Avg. | wp | review | blog | tweets | Avg. | |
| *Zero-shot Methods* | | | | | | | | | | |
| Likelihood | 0.3811 | 0.4342 | 0.4859 | 0.4337 | 0.5758 | 0.5665 | 0.6950 | 0.8730 | **0.6776** | 0.5731 |
| Rank | 0.3580 | 0.4544 | 0.4839 | 0.4321 | 0.5931 | 0.5768 | 0.8115 | 0.8402 | 0.7054 | **0.5883** |
| Logrank | 0.3831 | 0.4503 | 0.4937 | 0.4424 | 0.5754 | 0.5534 | 0.6852 | 0.8598 | 0.6685 | 0.5716 |
| Entropy | 0.5902 | 0.6273 | 0.4935 | **0.5703** | 0.4436 | 0.3869 | 0.3057 | 0.1648 | 0.3253 | 0.4303 |
| DetectGPT | 0.4778 | 0.4659 | 0.5452 | 0.4963 | 0.6129 | 0.5791 | 0.5546 | 0.6778 | 0.6061 | 0.5590 |
| Fast-DetectGPT | 0.4107 | 0.5753 | 0.5197 | 0.5019 | 0.6649 | 0.4945 | 0.6471 | 0.7409 | 0.6369 | 0.5790 |
| *Supervised Methods* | | | | | | | | | | |
| GLTR-based | 0.5180 | 0.5558 | 0.6033 | 0.5590 | 0.6768 | 0.6713 | 0.6861 | 0.6793 | 0.6784 | 0.6272 |
| PPL-based | 0.5707 | 0.6119 | 0.6641 | 0.6156 | 0.6260 | 0.7194 | 0.6387 | 0.7654 | 0.6874 | 0.6566 |
| Ghostbuster | 0.6021 | 0.5525 | 0.6179 | 0.5908 | 0.7270 | 0.6880 | 0.7488 | 0.6582 | 0.7055 | 0.6564 |
| Likelihood-based | 0.5239 | 0.5057 | 0.4964 | 0.5087 | 0.4996 | 0.5005 | 0.4942 | 0.4996 | 0.4985 | 0.5028 |
| Rank-based | 0.4967 | 0.5691 | 0.5014 | 0.5224 | 0.6441 | 0.5961 | 0.5282 | 0.6494 | 0.6045 | 0.5693 |
| Logrank-based | 0.4835 | 0.5438 | 0.5382 | 0.5218 | 0.6460 | 0.5827 | 0.4142 | 0.7880 | 0.6077 | 0.5709 |
| Entropy-based | 0.4466 | 0.5392 | 0.5154 | 0.5004 | 0.5476 | 0.5495 | 0.4828 | 0.6696 | 0.5624 | 0.5358 |
| XLNet | 0.9699 | 0.9290 | 0.9025 | 0.9338 | 0.9852 | 0.9802 | 0.9862 | 0.9767 | 0.9821 | 0.9614 |
| DeBERTa | 0.9798 | 0.9821 | 0.9384 | **0.9668** | 0.9941 | 0.9883 | 0.9912 | 0.9777 | **0.9878** | **0.9788** |
| RoBERTa | 0.9825 | 0.9646 | 0.7650 | 0.9040 | 0.7503 | 0.6358 | 0.9627 | 0.9933 | 0.8355 | 0.8649 |
| DistilBERT | 0.9433 | 0.9380 | 0.8934 | 0.9249 | 0.9765 | 0.9841 | 0.9827 | 0.9851 | 0.9821 | 0.9576 |
| XLM-RoBERTa | 0.9577 | 0.9471 | 0.8580 | 0.9209 | 0.9750 | 0.9859 | 0.9851 | 0.9923 | 0.9846 | 0.9573 |
| Mistral | 0.9816 | 0.9802 | 0.8375 | 0.9331 | 0.9668 | 0.9249 | 0.9668 | 0.3369 | 0.7989 | 0.8564 |
| Llama | 0.9549 | 0.9759 | 0.8867 | 0.9392 | 0.9162 | 0.9502 | 0.9885 | 0.7182 | 0.8933 | 0.9129 |
| ImBD | 0.8342 | 0.7526 | 0.7782 | 0.7883 | 0.9439 | 0.8059 | 0.9492 | 0.7849 | 0.8710 | 0.8356 |
| DPIC | 0.9091 | 0.8240 | 0.8253 | 0.8528 | 0.8793 | 0.8963 | 0.9233 | 0.8712 | 0.8925 | 0.8755 |
| DetectAnyLLM | 0.9543 | 0.8253 | 0.5635 | 0.7810 | 0.6074 | 0.7283 | 0.9472 | 0.8725 | 0.7889 | 0.7855 |
| MPU | 0.5247 | 0.5162 | 0.3858 | 0.4756 | 0.5003 | 0.4948 | 0.4914 | 0.5121 | 0.4996 | 0.4893 |
| Overall Avg. | 0.6764 | 0.6884 | 0.6497 | 0.6715 | 0.7220 | 0.7016 | 0.7444 | 0.7453 | **0.7283** | —— |

(b) Cross-Model (CM)

| AUC | Llama3.1-8b | DeepSeek-chat | gpt4o-mini | Qwen-plus | Avg. |
|---|---|---|---|---|---|
| CM Avg. | 0.7109 | 0.7676 | 0.7674 | **0.7688** | — |

(c) Cross-Operation (CO)

| AUC | Creative Reformation | | | | Linguistic Enhancement | | | | Cross-lingual | Overall Avg. |
|---|---|---|---|---|---|---|---|---|---|---|
| | create | rewrite | summary | Avg. | polish | refine | expand | Avg. | translate | |
| CO Avg. | 0.7798 | 0.7556 | 0.7242 | **0.7532** | 0.7510 | 0.7299 | 0.7776 | 0.7528 | 0.6960 | — |

## 5.2 Performance of LLM Grouping on Text Detectability: LLM-Va

**Model size has no significant impact on the detection difficulty.** We investigated how a model's size impacts the detectability of its generated text. In addition to the four baseline models, we conducted CM experiments using texts from GPT-4o, Qwen2-7B-Instruct, and Qwen3-8B. As shown in Table 3a, we found a surprising trend: while the average detection performance (AUROC of around 77.28%) for large models was slightly higher than for smaller models (AUROC of around 76.07%), the difference was minimal and likely not statistically significant. This finding suggests that contrary to expectations, *text generated by larger models does not exhibit significant resistance to detection*. We hypothesize this is because although larger models generate higher-quality text, it's not always flawless. Conversely, the inconsistent quality of text from smaller models can sometimes make it harder to distinguish from human-written content.

---

[1] https://gptzero.me

[2] https://www.zerogpt.com

[3] https://gowinston.ai

Table 3: The average detection performance (AUC) of detectors over LLM-Va experiments. Detailed results are available in Appendix C.4.

(a) Small vs Large

| AUC | Small | | | | Large | | |
|---|---|---|---|---|---|---|---|
| | gpt4o-mini | Qwen2-7b-instruct | Qwen3-8b | Avg. | gpt4o | Qwen-plus | Avg. |
| Overall Avg. | 0.7783 | 0.7572 | 0.7465 | 0.7607 | 0.7890 | 0.7565 | 0.7728 |

(b) Instruction vs Base

| AUC | Llama | | Qwen | |
|---|---|---|---|---|
| | Llama3.1-8b | Llama3.1-8b-instruct | Qwen3-8b | Qwen2-7b-instruct |
| Overall Avg. | 0.7272 | 0.6977 | 0.7465 | 0.7572 |

(c) Reasoning vs Non-Reasoning

| AUC | GPT | | | Qwen | | Deepseek | |
|---|---|---|---|---|---|---|---|
| | gpt4o-mini | gpt5 | gpt5-rs | Qwen-plus | Qwen-plus-rs | DeepSeek-chat | DeepSeek-rs |
| Overall Avg. | 0.7774 | 0.7407 | 0.7329 | 0.7594 | 0.7584 | 0.7551 | 0.7517 |

**Instruction strategy affects detection difficulty.** We then examine the influence of instruction tuning on the detectability of generated texts. As presented in the Table 3b, the results indicate that Llama3.1-8B-Instruct is marginally more challenging to detect than base model. In contrast, Qwen2-7B-Instruct is easier to detect compared to Qwen3-8B. We hypothesize that the difference may stem from the instruction tuning applied to these models. Specifically, the Llama3.1 instruction-tuned models are designed for multilingual dialogue applications, which likely enhances their ability to generate outputs that more closely resemble human-written text, thereby making them less distinguishable by detection systems. This finding suggests that a model's instruction strategy plays a crucial role in determining the difficulty of detecting its generated texts.

**Reasoning-enhanced texts are harder to detect.** Enhanced reasoning capabilities have become a major focus in the development of large language models, with most state-of-the-art systems now deeply integrating advanced reasoning and chain-of-thought functionalities. To investigate the impact of such reasoning-enhanced modes on the detectability of AI-generated text, we conducted a comparative analysis using GPT-5, Qwen-Plus, and DeepSeek, evaluating texts generated in reasoning mode vs non-reasoning mode (Table 3c). Overall, the text generated in the reasoning mode is more difficult to detect. This might be because the reasoning model conducts extensive thinking before generating, eliminating machine-like expressions. Therefore, the text produced by the reasoning model often has a less regular structure and is closer to human language, thereby increasing the difficulty of detection.

## 5.3 IMPACT OF TEXT GENERATE COMBINATION: LLM/OPERATION-CROSSOVER

### 5.3.1 LLM-CROSSOVER

**Secondary generation generally reduces detectability.** For most LLM combinations, secondary generation decreases the detectability of AI-generated texts. Results shown in Figure 2(b) indicate that, detection difficulty tends to decrease after reprocessing, particularly for texts generated by GPT-4o, DeepSeek, and Qwen. This suggests that secondary generations often introduce variations or less predictable patterns, making the texts more human-like and harder to detect.

**Llama as the second-generation model can increase detectability.** When Llama is used to reprocess texts initially generated by DeepSeek, GPT-4o, or Qwen, detection performance often improves. In these cases, the stylistic consistency between Llama and the initial models may produce higher-quality secondary generations that are easier to distinguish from human text. Conversely, greater divergence between other model combinations tends to reduce output quality, increasing detection difficulty.

### 5.3.2 OPERATION-CROSSOVER

As shown in Figure 2(c), not all operation combinations increase detection difficulty. In particular,

**Supervised detectors are more robust to operation crossover.** For most of the 17 operation combinations, supervised methods experienced reduced performance. This can be attributed to their reliance on training data, as combined operations amplify the distribution shift between training and test samples, thereby lowering detection accuracy.

**Certain operation combinations improve zero-shot detection.** In contrast to supervised methods, some zero-shot detectors showed enhanced performance when applied to combinations such as translate–expand and translate–rewrite. We hypothesize that the first operation produces text that is inherently difficult to detect, while the second introduces recognizable features that improve overall detectability.

These findings indicate that secondary operations do not always hinder detection. In some cases, they may even facilitate it, offering potential directions for designing more effective detection strategies. Detailed results are shown in Appendix C.5.

### 5.4 SUGGESTIONS FOR BETTER LLM-GENERATED TEXT DETECTORS CONSTRUCTION

#### 5.4.1 ZERO-SHOT DETECTOR RECOMMENDATION

**Given the features of target data, how can we select the most suitable zero-shot detector?** Based on the results in Table 2a, C.1 and C.2, we summarize the following suggestions: 1) For formal texts, entropy-based methods perform best, while for informal texts, log-rank yields the highest accuracy. 2) For content generated by Llama3.1-8B, entropy remains most effective, whereas for other LLM series, Fast-DetectGPT outperforms other methods. 3) In creative reformulation and translation scenarios, Fast-DetectGPT shows the highest robustness, while linguistic enhancements are better handled by Rank.

#### 5.4.2 SUPERVISED DETECTOR TRAINING

**What is the best training condition?** Since supervised models show the best performance across prior evaluations, we further investigate how to effectively train a supervised detector. We conduct a systematic analysis of embedding-based detectors under different training configurations (Appendix C.6). We found that: 1) For training epochs, with a training set of approximately 4k samples, encoder-based models reach peak performance within 2–3 epochs, demonstrating fast convergence. In contrast, decoder-based models such as Mistral benefit from slightly more epochs for stable results. 2) For the loss function, the optimal range is when the loss is below 0.01, striking a balance between training efficiency and model performance. Setting overly strict thresholds may lead to overfitting.

**What is the best training scale?** To explore the impact of data scale, we conducted controlled experiments by reducing training set sizes to 2,000, 1,000, and 500 samples. All models were evaluated under CD/CM/CO setups. As shown in Figure 3, DeBERTa consistently achieves the highest and most stable performance across all scales, demonstrating strong generalization. Notably, training with more than 4,000 samples may lead to overfitting in some models. We therefore recommend limiting training to 2,000–3,000 samples when test sets contain around 8,000 instances. Across all settings, supervised methods maintain a substantial advantage over zero-shot baselines, confirming their robustness for detection tasks.

#### 5.4.3 OPEN-ACCESS DETECTOR SELECTION

To assess generalization and detection performance, we evaluated five open-access detectors on the BRED benchmark: RADAR, OpenAI's RoBERTa-based detector, RoBERTa-HC3, ArguGPT, and MPU. As shown in Figure 4, *ArguGPT achieves the highest average AUC across all CD/CM/CO settings*, demonstrating strong generalizability across diverse text types. RADAR also performs competitively in certain scenarios. In contrast, OpenAI's RoBERTa-based detector performs consistently poorly, likely due to its design focus on detecting GPT-2 outputs. Based on these results, we recommend **ArguGPT as the most effective open-access detector, with MPU and RADAR**

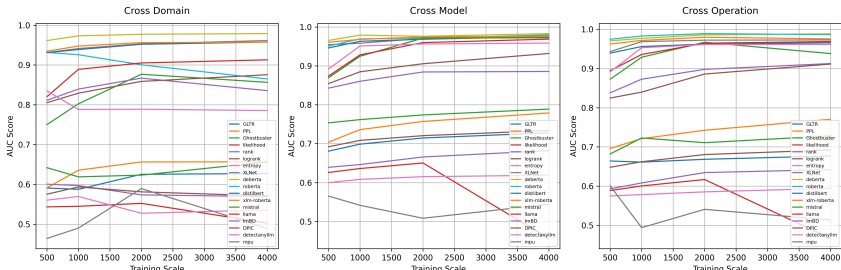

Figure 3: The average performance (AUC) of supervised detectors across different scales of training data.

**as reasonable alternatives depending on the application**. Full results and additional analysis are provided in Appendix C.7.

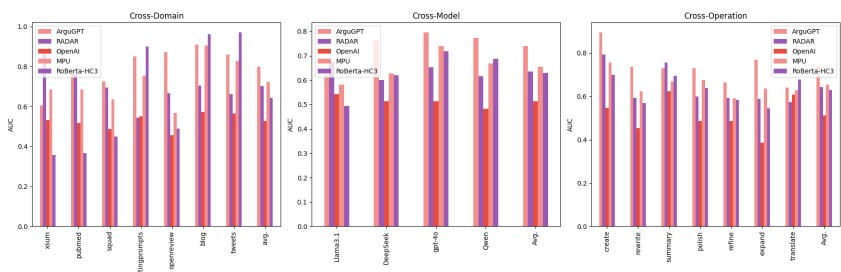

Figure 4: The average performance (AUC) of open-access detectors over CD/CM/CO experiments.

### 5.4.4 COMMERCIAL DETECTOR COMPARISON

Table 4: The average performance (AUC) of commercial detectors over CD/CM/CO experiments.

| AUC | Cross-Domain | Cross-Model | Cross-Operation |
|---------|--------------|-------------|-----------------|
| ZeroGPT | 0.7114 | 0.6893 | 0.7023 |
| GPTZero | 0.8908 | 0.8838 | 0.8854 |
| Winston | **0.9070** | **0.9050** | **0.9101** |

We evaluated three commercial detectors (GPTZero, Zero-GPT, and Winston AI) on various text types, with 100 sampled texts per domain/model/operation due to monthly request limits. Results in Table 4 show that Winston AI outperformed the others, achieving AUROC scores of 90.7%, 90.5%, and 91.01% in CD/CM/CO tasks. **GPTZero also performed well, but slightly worse than Winston AI, while Zero-GPT showed lower accuracy.**

## 6 CONCLUSION AND LIMITATION

In this work, we introduce BRED, a comprehensive benchmark for evaluating the detection of LLM-generated text across various domains, operations, and model architectures. Through a large-scale, multi-faceted dataset and extensive empirical analysis, BRED facilitates systematic comparisons and provides valuable insights for advancing AI-generated text detection.

BRED has two key limitations. Firstly, its current scale is primarily suited for small- to medium-sized applications, and the evaluation of larger datasets is necessary for validating industrial use. Secondly, it excludes watermark-based detectors due to incompatible evaluation protocols. These constraints highlight areas for further development.

# 7 ETHICS STATEMENT

All data used in this study are sourced from publicly available datasets with appropriate licenses. Any personal information or potentially harmful content present in the datasets has been carefully identified and removed through a combination of automated filtering and manual review. We have ensured that our experiments comply with ethical standards for research and data privacy, and no sensitive or private information is included in our analysis.

# 8 REPRODUCIBILITY STATEMENT

To ensure the reproducibility of this research, all data, code, and models used in this study are publicly available. The methodologies and experiments are fully documented, and the necessary instructions to replicate the results can be found in the supplementary materials. All relevant resources, including datasets and code implementations, can be accessed via https://anonymous.4open.science/r/BRED-65D3.

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

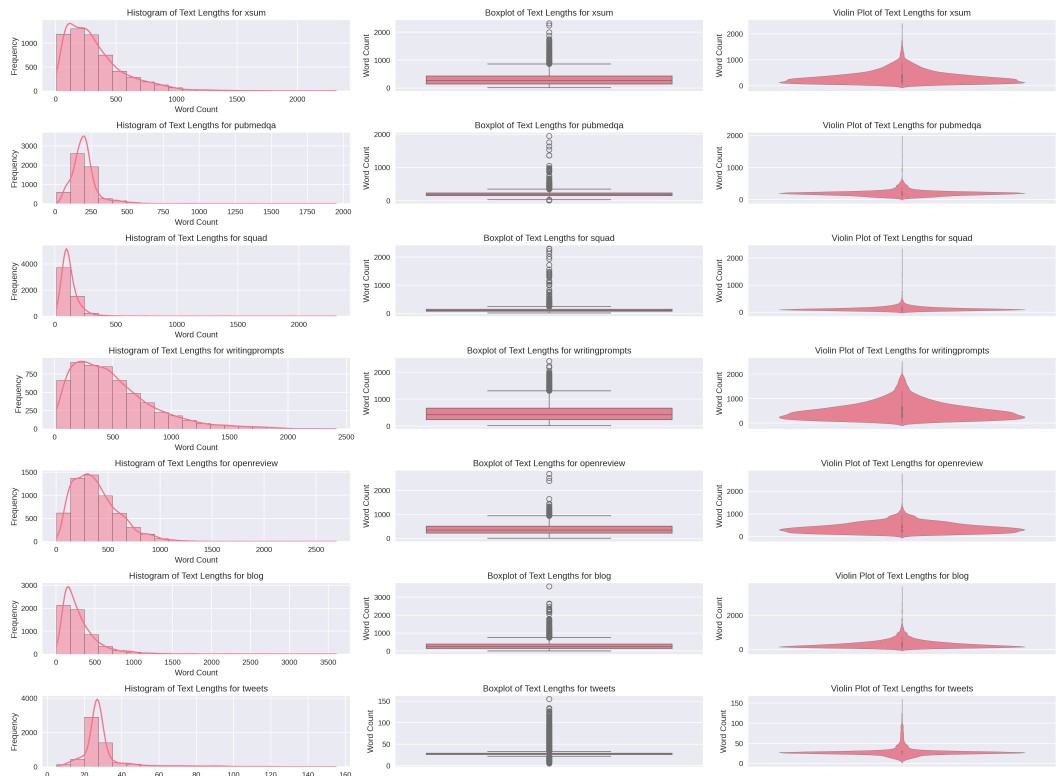

Figure B.1: Data distribution cross different domains.

# A    USE OF LLMs

LLMs were employed for two main tasks in this research: article polishing and code modification. For article polishing, LLMs enhanced clarity, grammar, and style, improving the overall quality of the text. In code modification, LLMs identified errors, optimized code, and suggested improvements, streamlining the debugging and development process. These applications significantly improved efficiency and output quality.

# B    DETAILED INFORMATION OF DATASET

## B.1    DATASET DESCRIPTIONS

This section provides detailed descriptions of the datasets used across various tasks in BRED, outlining the specific data sources and their characteristics. The collected data underwent a cleaning process, which included the following steps: 1) removal of escape characters in the text; 2) elimination of sections containing non-English text; 3) removal of obvious LLM prompt phrases such as "Here is the translation" and "note."

## B.2    HUMAN-WRITTEN TEXT COLLECTION

**Xsum**    Our news dataset is based on the XSum(Narayan et al., 2018) dataset, which consists of BBC articles and ac companying single sentence summaries. The dataset contains 226,711 BBC articles archived via the Wayback Machine, spanning nearly a decade (from 2010 to 2017) and covering a wide range of domains, including news, politics, sports, weather, business, technology, science, health, family, education, entertainment, and the arts.

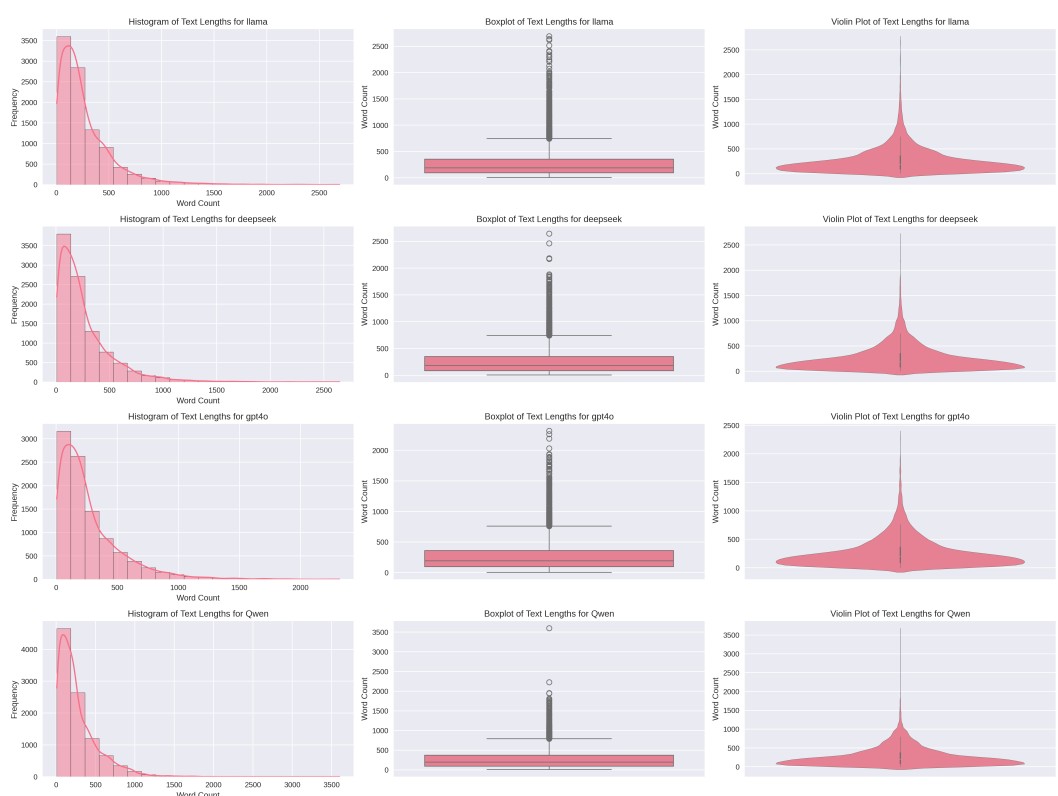

Figure B.2: Data distribution cross different LLMs.

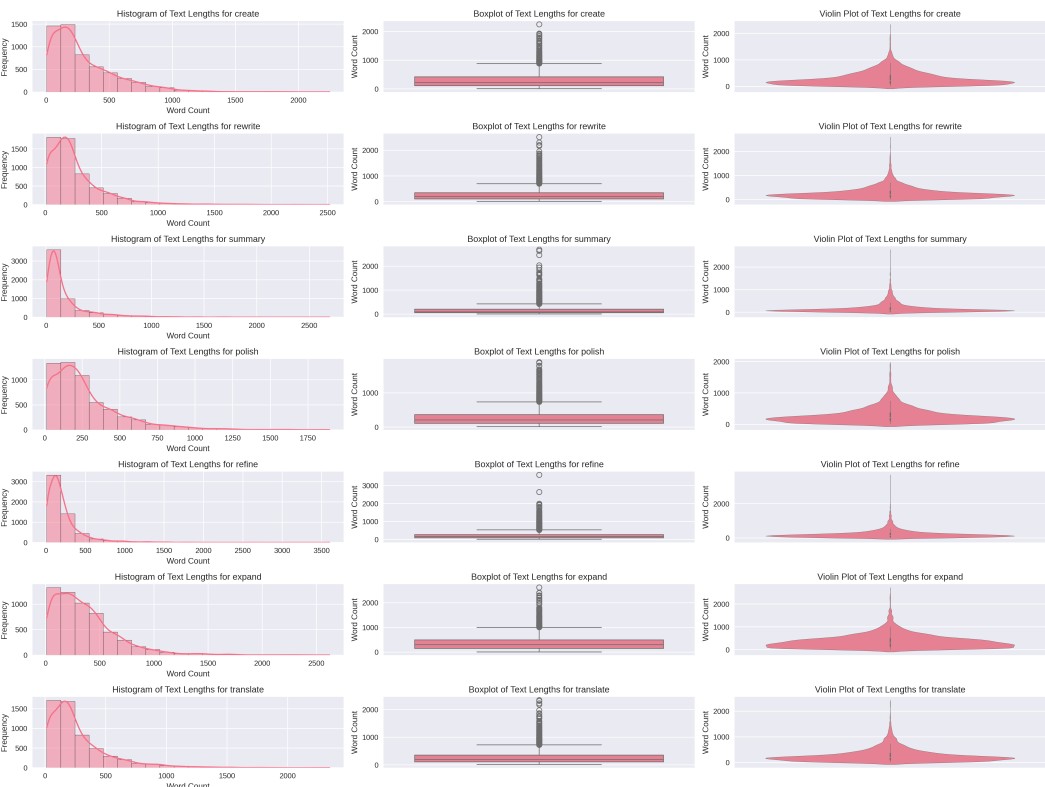

Figure B.3: Data distribution cross different operations.

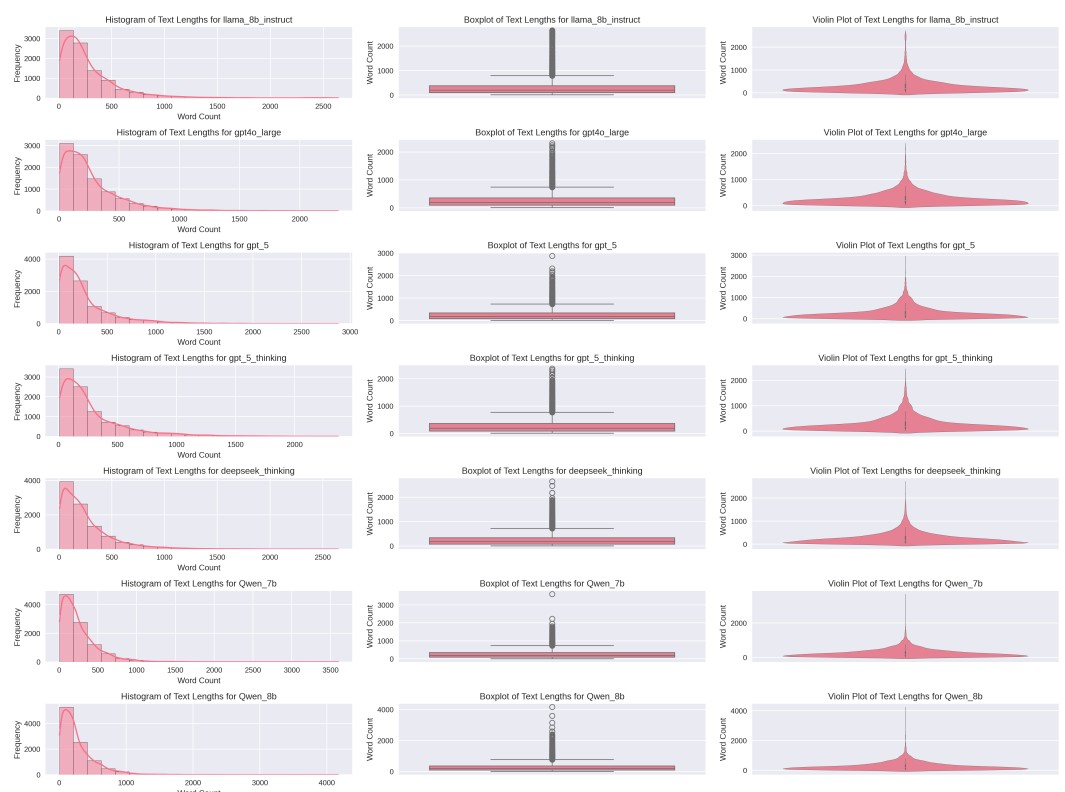

Figure B.4: Data distribution cross different LLM variants in LLM-Va tasks.

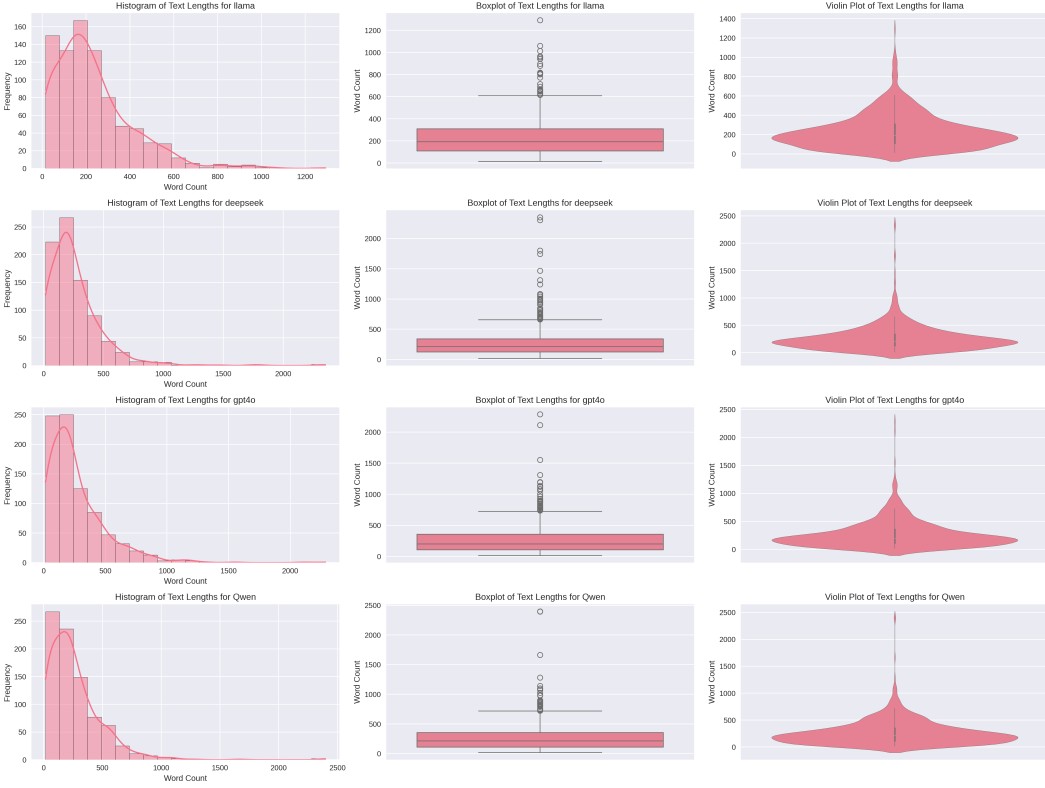

Figure B.5: Data distribution cross different LLM combinations in LLM-Co task.

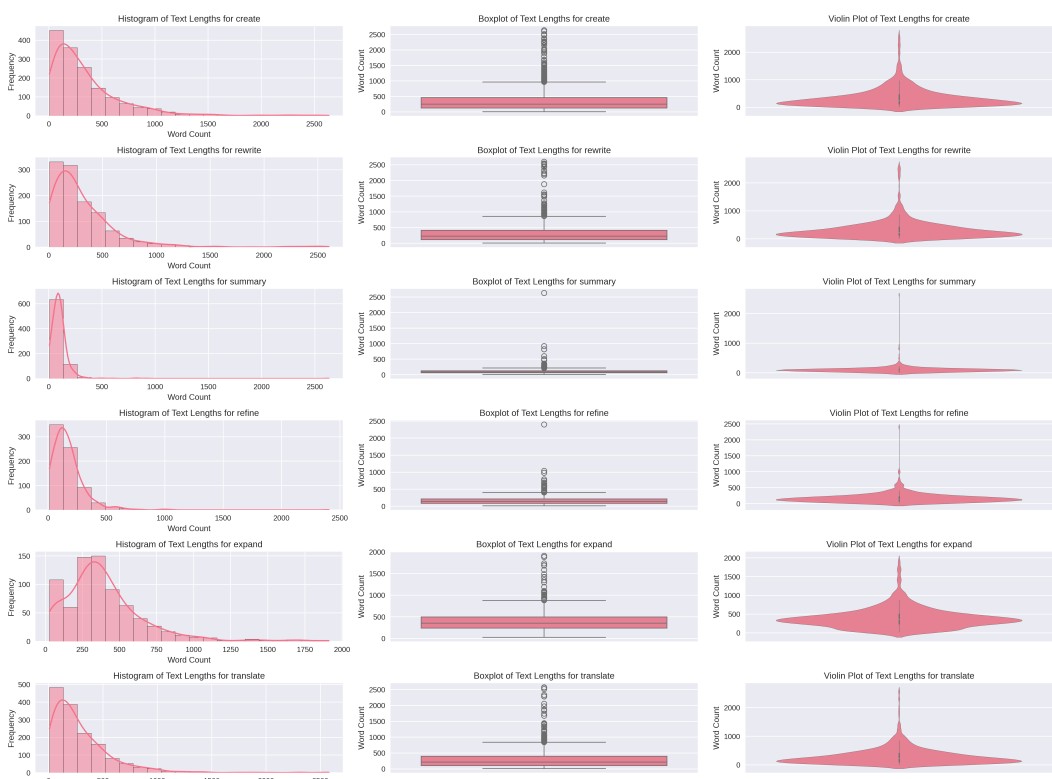

Figure B.6: Data distribution cross different operation combinations in Op-Co task.

**Wp**  Our creative writing dataset is constructed based on the WritingPrompts(Fan et al., 2018) dataset, which comprises prompts and corresponding stories collected from the Reddit's r/WritingPrompts[1] before 2018. The dataset has been preprocessed to remove noise, with story lengths constrained to a minimum of 30 words and a maximum of 1,000 words.

**Pubmed**  Our academic article data is sourced from the publicly available PubMedQA dataset(Jin et al., 2019), which is constructed from PubMed[2] abstracts and includes abstract samples along with human-generated question-answer pairs. In accordance with the generation operation defined in this study, we primarily selected abstracts from the PubMedQA dataset for data collection.

**Squad**  Our Wikipedia data is sourced from the publicly available SQuAD(Rajpurkar et al., 2016; 2018) dataset released by Stanford University, which is a reading comprehension dataset comprising over 100,000 questions based on a set of Wikipedia articles. Similar to our approach with PubMedQA, we selected the context as part of the human-written text in our data collection process.

**Review**  Recent studies have documented a growing trend of peer reviewers employing LLMs to generate or refine review comments. To account for this phenomenon in our dataset construction, we collected peer reviews from Openreview[3], limiting our selection to those published before 2019 in order to minimize potential LLM involvement in the generation of the comments.

**Blog**  To enrich the variety of text types in our dataset, we also collected blog articles, which are sourced from a publicly available dataset(Schler et al., 2006), consisting of posts from 19,320 bloggers gathered from Blogger website[4] in August 2004.

---

[1]https://www.reddit.com/r/WritingPrompts/

[2]https://pubmed.ncbi.nlm.nih.gov/

[3]https://openreview.net/

[4]https://www.blogger.com/

**Tweets** Finally, considering the challenges of short-text detection in the current field of LLM-generated text detection, we collected 2,800 tweets from Twitter dataset(Go et al., 2009). After data cleaning, the average text length is approximately 30 words.

### B.3 LLM-based text operations

**create** The 'create' operation of LLMs involves generating new content based on user prompts. Specifically, we adopt complete operation, which the first 25 percent of a human-written text is used as input, and the LLMs are prompted to complete the remaining parts based on the initial fragment.

**rewrite** The 'rewrite' operation involves prompting a large language model (LLM) with an existing text, instructing it to rephrase the words and reorganize sentences or paragraphs to change the writing style, producing a new text that preserves the original meaning while altering its linguistic form. This operation is applied at the document level, often resulting in structural changes to the original text.

**summary** The 'summary' operation refers to the process in which large language models condense and rephrase the original text after receiving it, extracting the core ideas and key information to produce a clear and concise summary, similar to the process of summarizing an abstract. The text produced after the summary operation typically does not retain the original content.

**refine** The 'refine' operation involves simplifying, condensing, or improving the text after receiving it, making it more concise, clear, and easier to comprehend. This operation is applied at the sentence level, where the refined text retains some parts of the original content but presents it in a more concise manner.

**polish** The 'polish' operation refers to the process by which a LLM enhances the sentence-level quality of a given text while preserving its original content and meaning. This operation typically involves correcting grammatical errors, refining word choices, improving sentence fluency, and ensuring stylistic consistency. Unlike rewrite or refine which may restructure or compress the input, the polish operation maintains the original sentence structure and focuses only on linguistic clarity and readability.

**expand** The 'expand' operation denotes the procedure through which a LLM adds relevant content to a given text, such as additional explanations, examples, or background information. While keeping the original meaning intact, the model makes the text more detailed and informative. This operation is also performed at the sentence level and does not alter the overall structure of the original text.

**translate** The 'translate' operation denotes the sequence-to-sequence transformation process in which a language an LLM converts a text from one language to another while preserving the original meaning, tone, and style as much as possible. In this study, since our dataset is composed of English texts, we apply the translate operation in two stages. First, each human-written text is translated from English to Chinese, and then translated back into English.

### B.4 Prompt templates for text generation

The example prompts we used in the generation of LLM texts are shown below in Table B.1.

### B.5 Experimental Setup

We define several tasks for performance evaluation, including 1) Cross-Domain (CD) experiment, Cross-Model (CM) experiment, Cross-Operation (CO) experiment and Multi-Length (ML) experiment, 2) LLM-Variant (LLM-Va) experiment, LLM-Crossover (LLM-Co), and 3) Operation-Crossover (Op-Co) experiment. In particular, the detailed train/test splits of the three categories of tasks are shown below in Table B.2, B.3, and B.4. As for ML experiment, we mainly evaluate the cross-domain performance across various detectors, which settings are the same as that in CD experiment. Notably, among the detectors evaluated in our experiments, the original implementation

Table B.1: Prompt templates for text generation.

| Prompt Part | Description | Example |
|---|---|---|
| **Part I** | This section defines the role and context of the task. It clarifies the type of operation to be performed. | "You are a (eg. news writer), tasked with improving and continuing a [text type]. " |
| **Part II** | This section defines the operation you are performing. This part will be one of the following: 1) create, 2) rewrite, 3) summary, 4) refine, 5) polish, 6) expand, 7) translate. Each option specifies the type of modification to the text. | **1) Create**: "Your task is to continue the article in a consistent tone, adding depth and seamlessly transitioning from the provided content..." 
 **2) Rewrite**: "Text rewriting refers to the process in which, upon receiving a text input, the language model adjusts word choices..." 
 **3) Summary**: "Text summarization refers to the process where, after inputting text, the language model generates a concise and accurate abstract..." 
 **4) Refine**: "Text refining refers to the process where, after inputting text, the language model simplifies, compresses, or rewrites the given content into a version that is more concise..." 
 **5) Polish**: "Text polishing refers to the process where, after inputting text, the model improves the quality, fluency..." 
 **6) Expand**: "Text expansion refers to the process where, given an input text, the language model generates new content..." 
 **7) Translate**: "Translate the article into [desired language], ensuring the meaning and tone are preserved." |
| **Part III** | This section includes the provided content that needs to be worked on the given text. | "The given text is as follows: [insert the original article or content]" |
| **Part IV** | This section contains any additional notes, instructions, or specific clarifications for the task. | "Note: Your rewriting needs to conform to the standards and expressions of [text type]." |

Table B.2: Train/ Test dataset splits for Cross-Domain (CD), Cross-Model (CM), and Cross-Operation (CO) Experiments.

| Tasks | Datasets | Human / AI texts | | Domains(train) | LLMs(train) | Operation(train) |
| --- | --- | --- | --- | --- | --- | --- |
| | | train | test | | | |
| CD | xsum wp pubmed squad review blog tweets | 2k/2k | 2.8k/2.8k | w/o xsum w/o wp w/o pubmed w/o squad w/o review w/o blog w/o tweets | Llama3.1-8b-v2, DeepSeek, gpt-4o-mini, Qwen-plus | create, translate, polish, expand, refine, summary, rewrite |
| CM | Llama-3.1 DeepSeek gpt4o-mini Qwen-plus | 2k/2k | 4.9k/4.9k | xsum, wp, pubmed, squad, review, blog, tweets | w/o Llama-3.1-8b w/o DeepSeek w/o gpt-4o w/o Qwen-plus | create, translate, polish, expand, refine, summary, rewrite |
| CO | create translate polish expand refine summary rewrite | 2k/2k | 2.8k/2.8k | xsum, wp, pubmed, squad, review, blog, tweets | Llama3.1-8b-v2, DeepSeek, gpt-4o-mini, Qwen-plus | w/o create w/o translate w/o polish w/o expand w/o refine w/o summary w/o rewrite |

Table B.3: Train/ Test dataset splits for LLM-Va task.

| Tasks | Datasets | Human / AI texts | | Domains(train) | LLMs(train) | Operation(train) |
| --- | --- | --- | --- | --- | --- | --- |
| | | train | test | | | |
| small v.s. large | gpt4o-mini gpt-4o Qwen2-7b-instruct Qwen3-8b Qwen-plus | 2k/2k | 4.9k/4.9k | xsum, wp, pubmed, squad, review, blog, tweets | w/o GPT series w/o Qwen series | create, translate, polish, expand, refine, summary, rewrite |
| instruct v.s. base | Llama3.1-8b Llama3.1-8b-instruct Qwen2-7b-instruct Qwen3-8b | 2k/2k | 4.9k/4.9k | xsum, wp, pubmed, squad, review, blog, tweets | w/o Llama series w/o Qwen series | create, translate, polish, expand, refine, summary, rewrite |
| reasoning | gpt4o-mini gpt-5 gpt-5(reasoning) Qwen-plus Qwen-plus(reasoning) DeepSeek-chat DeepSeek-reasoning | 2k/2k | 4.9k/4.9k | xsum, wp, pubmed, squad, review, blog, tweets | w/o GPT series w/o Qwen series w/o DeepSeek series | create, translate, polish, expand, refine, summary, rewrite |

of DPIC(Yu et al., 2024) was unavailable. We therefore conducted performance comparisons using our reproduced version based on the method description in the original paper.

Table B.4: Train/ Test dataset splits for LLM-Co and Op-Co tasks.

| Tasks | Datasets | Human / AI texts | | Domains(train) | LLMs(train) | Operation(train) |
| --- | --- | --- | --- | --- | --- | --- |
| | | train | test | | | |
| LLM-Co | Llama3.1-8b DeepSeek gpt4o-mini Qwen-plus | 169/169 166/166 167/167 169/169 | 0.2 | xsum, wp, pubmed, squad, review, blog, tweets | Llama3.1-8b DeepSeek gpt4o-mini Qwen-plus | translate, polish, rewrite |
| Op-Co | create rewrite summary refine expand translate | 305/305 226/226 153/153 150/150 148/148 296/296 | 0.2 | xsum, wp, pubmed, squad, review, blog, tweets | Llama3.1-8b, DeepSeek, gpt4o-mini, Qwen-plus | create rewrite summary refine expand translate |

Table C.1: The detection performance (AUC) of various detectors in Cross-Model (CM) experiment.

| AUC | Llama3.1-8b | DeepSeek-chat | gpt4o-mini | Qwen-plus | avg. |
|---|---|---|---|---|---|
| *Zero-shot Methods* | | | | | |
| Likelihood | 0.399 | 0.5744 | 0.5679 | 0.5854 | 0.5316 |
| Rank | 0.5089 | 0.5782 | 0.5786 | 0.5844 | 0.5625 |
| Logrank | 0.4417 | 0.5633 | 0.5642 | 0.5787 | 0.5369 |
| Entropy | 0.5553 | 0.4377 | 0.4451 | 0.4152 | 0.4633 |
| DetectGPT | 0.4328 | 0.4846 | 0.51 | 0.5114 | 0.4847 |
| Fast-DetectGPT | 0.3936 | 0.6143 | 0.663 | 0.5996 | **0.5676** |
| *Supervised Methods* | | | | | |
| GLTR-based | 0.6865 | 0.7371 | 0.7459 | 0.7418 | 0.7278 |
| PPL-based | 0.7349 | 0.7853 | 0.8086 | 0.7874 | 0.7790 |
| Ghostbuster | 0.7158 | 0.8399 | 0.7768 | 0.8235 | 0.7890 |
| Likelihood-based | 0.4438 | 0.5 | 0.4424 | 0.4389 | 0.4562 |
| Rank-based | 0.6919 | 0.6667 | 0.6862 | 0.6838 | 0.6821 |
| Logrank-based | 0.7477 | 0.7194 | 0.736 | 0.7328 | 0.7339 |
| Entropy-based | 0.6049 | 0.6228 | 0.6219 | 0.6262 | 0.6189 |
| XLNet | 0.9466 | 0.9952 | 0.9931 | 0.9843 | 0.9798 |
| DeBERTa | 0.9482 | 0.9983 | 0.998 | 0.9881 | **0.9831** |
| RoBERTa | 0.9317 | 0.9968 | 0.9971 | 0.9889 | 0.9786 |
| DistilBERT | 0.9508 | 0.9893 | 0.9865 | 0.9753 | 0.9754 |
| XLM-RoBERTa | 0.9504 | 0.9945 | 0.9917 | 0.9685 | 0.9762 |
| Mistral | 0.9596 | 0.9764 | 0.9855 | 0.9685 | 0.9725 |
| Llama | 0.9383 | 0.9854 | 0.9831 | 0.9692 | 0.969 |
| ImBD | 0.7436 | 0.938 | 0.9346 | 0.9271 | 0.8858 |
| DPIC | 0.9153 | 0.9504 | 0.9277 | 0.9352 | 0.9321 |
| DetectAnyLLM | 0.9088 | 0.9765 | 0.9804 | 0.9697 | 0.9588 |
| MPU | 0.5138 | 0.4999 | 0.4951 | 0.6681 | 0.5442 |
| Overall Avg. | 0.7109 | 0.7676 | 0.7674 | **0.7688** | — |

# C  DETAILED RESULTS OF ABLATION EXPERIMENTS

## C.1  EVALUATION PERFORMANCE (AUC) OF OOD EXPERIMENTS

Table C.1 and  C.2 shows the full results of detection performance the OOD Experiments: Cross-Model (CM), and Cross-Operation (CO).

## C.2  EVALUATION PERFORMANCE (AUPR) OF OOD EXPERIMENTS

To evaluate the model's detection capability, we employed the Area Under the ROC Curve (AUC) as the primary metric in our experiments (CD/CM/CO). Considering potential class imbalance, we additionally report the Area Under the PR Curve (AUPR) to provide a more comprehensive performance assessment. Detailed information can be found in Table C.3, C.4, and C.5.

## C.3  EVALUATION PERFORMANCE (AUC) OF ML EXPERIMENT

We created a multi-length dataset by segmenting the texts into fragments of different sizes (10, 50, 100, 200, 500 words). This task provides a practical framework for evaluating model robustness across varying text lengths. The detailed results are shown below in Table C.6,  C.7, and  C.8.

## C.4  EVALUATION PERFORMANCE (AUC) OF LLM-VA

We reported the performance on LLM-Va experiments, the results are shown in Table C.9, C.10, C.11.

## C.5  EVALUATION PERFORMANCE (AUC) OF LLM-CO AND OP-CO

To explore the influence of the combinations of different LLMs or different operations during the process of text generation, we did LLM-Co and Op-Co experiments in this paper. Detailed settings can be found in section3 and Appendix. The detailed results are shown below in Table C.12,  C.13, C.14,  C.15, and  C.16.

Table C.2: The detection performance (AUC) of various detectors in Cross-Operation (CO) experiment, grouped by operation types.

| AUC | Creative Reformation | | | | Linguistic Enhancement | | | | Cross-lingual | Overall Avg. |
|---|---|---|---|---|---|---|---|---|---|---|
| | create | rewrite | summary | Avg. | polish | refine | expand | Avg. | translate | |
| *Zero-shot Methods* | | | | | | | | | | |
| Likelihood | 0.7556 | 0.5118 | 0.3384 | 0.5353 | 0.5435 | 0.4316 | 0.6543 | 0.5431 | 0.4854 | 0.5315 |
| Rank | 0.6071 | 0.5886 | 0.4136 | 0.5364 | 0.5718 | 0.4874 | 0.7030 | **0.5874** | 0.5636 | 0.5622 |
| Logrank | 0.7746 | 0.5133 | 0.3310 | 0.5396 | 0.5452 | 0.4295 | 0.6712 | 0.5486 | 0.4910 | 0.5365 |
| Entropy | 0.2979 | 0.4667 | 0.6249 | 0.4632 | 0.4508 | 0.5302 | 0.3589 | 0.4466 | 0.5142 | 0.4634 |
| DetectGPT | 0.6177 | 0.5734 | 0.3765 | 0.5225 | 0.5121 | 0.5529 | 0.6660 | 0.5770 | 0.5541 | 0.5504 |
| Fast-DetectGPT | 0.8901 | 0.5053 | 0.3326 | **0.5760** | 0.5922 | 0.3718 | 0.7098 | 0.5579 | 0.5709 | **0.5675** |
| *Supervised Methods* | | | | | | | | | | |
| GLTR-based | 0.6978 | 0.6842 | 0.6864 | 0.6895 | 0.6877 | 0.6033 | 0.7280 | 0.6730 | 0.6516 | 0.6770 |
| PPL-based | 0.8101 | 0.7689 | 0.7949 | 0.7913 | 0.7583 | 0.7344 | 0.8080 | 0.7669 | 0.7200 | 0.7707 |
| Ghostbuster | 0.7612 | 0.7869 | 0.6634 | 0.7372 | 0.7252 | 0.6985 | 0.7478 | 0.7238 | 0.7147 | 0.7282 |
| Likelihood-based | 0.4325 | 0.4289 | 0.4771 | 0.4462 | 0.4276 | 0.5000 | 0.5005 | 0.4760 | 0.4996 | 0.4666 |
| Rank-based | 0.5798 | 0.6551 | 0.6941 | 0.6430 | 0.6503 | 0.6446 | 0.6229 | 0.6393 | 0.6478 | 0.6421 |
| Logrank-based | 0.5740 | 0.7105 | 0.7867 | 0.6904 | 0.7098 | 0.7199 | 0.6289 | 0.6862 | 0.7160 | 0.6923 |
| Entropy-based | 0.5964 | 0.6159 | 0.5595 | 0.5906 | 0.6187 | 0.5903 | 0.5978 | 0.6023 | 0.5814 | 0.5943 |
| XLNet | 0.9953 | 0.9942 | 0.9931 | 0.9953 | 0.9941 | 0.9910 | 0.9871 | 0.9907 | 0.8548 | 0.9728 |
| DeBERTa | 0.9992 | 0.9995 | 0.9968 | **0.9985** | 0.9976 | 0.9972 | 0.9969 | 0.9972 | 0.9316 | **0.9884** |
| RoBERTa | 0.9982 | 0.9981 | 0.9960 | 0.9974 | 0.9988 | 0.9984 | 0.9952 | **0.9975** | 0.9197 | 0.9863 |
| DistilBERT | 0.9968 | 0.9952 | 0.9846 | 0.9922 | 0.9807 | 0.9850 | 0.9799 | 0.9819 | 0.8363 | 0.9655 |
| XLM-RoBERTa | 0.9975 | 0.9964 | 0.9926 | 0.9955 | 0.9919 | 0.9943 | 0.9928 | 0.9930 | 0.8629 | 0.9755 |
| Mistral | 0.9236 | 0.9196 | 0.9154 | 0.9195 | 0.9691 | 0.9835 | 0.9853 | 0.9793 | 0.8699 | 0.9381 |
| Llama | 0.9957 | 0.9928 | 0.9922 | 0.9936 | 0.9840 | 0.9844 | 0.9913 | 0.9866 | 0.8407 | 0.9687 |
| ImBD | 0.9929 | 0.9733 | 0.9217 | 0.9626 | 0.9347 | 0.8672 | 0.9053 | 0.9024 | 0.7927 | 0.9125 |
| DPIC | 0.9282 | 0.9599 | 0.9340 | 0.9407 | 0.9051 | 0.9398 | 0.9382 | 0.9277 | 0.7739 | 0.9113 |
| DetectAnyLLM | 0.9890 | 0.9879 | 0.9622 | 0.9797 | 0.9871 | 0.9770 | 0.9903 | 0.9848 | 0.8361 | 0.9614 |
| MPU | 0.5035 | 0.5088 | 0.6125 | 0.5416 | 0.4869 | 0.5045 | 0.5040 | 0.4985 | 0.4758 | 0.5137 |
| Overall Avg. | **0.7798** | 0.7556 | 0.7242 | **0.7532** | 0.7510 | 0.7299 | 0.7776 | 0.7528 | 0.6960 | — |

Table C.3: The detection performance (AUPR) of various detectors in Cross-Domain (CD) experiment, grouped by formality levels (Formal vs. Informal) with overall average.

| AUPR | Formal | | | | Informal | | | | | Overall Avg. |
|---|---|---|---|---|---|---|---|---|---|---|
| | xsum | pubmed | squad | Avg. | wp | review | blog | tweets | Avg. | |
| *Zero-shot Methods* | | | | | | | | | | |
| Likelihood | 0.4485 | 0.4814 | 0.4874 | 0.4724 | 0.6335 | 0.6679 | 0.7013 | 0.8861 | 0.7222 | 0.6152 |
| Rank | 0.4177 | 0.4963 | 0.4985 | 0.4708 | 0.6490 | 0.6470 | 0.8390 | 0.8482 | 0.7458 | 0.6280 |
| Logrank | 0.4542 | 0.4935 | 0.4934 | 0.4804 | 0.6420 | 0.6583 | 0.7006 | 0.8749 | 0.7189 | 0.6167 |
| Entropy | 0.6389 | 0.6457 | 0.5088 | 0.5978 | 0.4901 | 0.4860 | 0.3875 | 0.3336 | 0.4243 | 0.4987 |
| DetectGPT | 0.4617 | 0.5183 | 0.5675 | 0.5158 | 0.6191 | 0.5983 | 0.5392 | 0.7037 | 0.6151 | 0.5725 |
| Fast-DetectGPT | 0.5304 | 0.6537 | 0.5946 | 0.5929 | 0.7344 | 0.6015 | 0.7062 | 0.7565 | 0.6997 | 0.6539 |
| *Supervised Methods* | | | | | | | | | | |
| GLTR-based | 0.5176 | 0.5629 | 0.5931 | 0.5579 | 0.6975 | 0.7093 | 0.7038 | 0.6964 | 0.7018 | 0.6401 |
| PPL-based | 0.5735 | 0.6043 | 0.6160 | 0.5979 | 0.6720 | 0.7291 | 0.7340 | 0.7787 | 0.7285 | 0.6725 |
| Ghostbuster | 0.5841 | 0.5746 | 0.5951 | 0.5846 | 0.6893 | 0.6891 | 0.7185 | 0.6331 | 0.6825 | 0.6405 |
| Likelihood-based | 0.4758 | 0.7441 | 0.4677 | 0.5625 | 0.3746 | 0.6503 | 0.3886 | 0.7497 | 0.5408 | 0.5501 |
| Rank-based | 0.5106 | 0.5743 | 0.4981 | 0.5277 | 0.6411 | 0.6140 | 0.4984 | 0.6993 | 0.6132 | 0.5765 |
| Logrank-based | 0.4979 | 0.5477 | 0.5502 | 0.5319 | 0.6307 | 0.6089 | 0.4235 | 0.7168 | 0.5950 | 0.5680 |
| Entropy-based | 0.4852 | 0.5167 | 0.5123 | 0.5047 | 0.5746 | 0.5844 | 0.4806 | 0.6411 | 0.5702 | 0.5421 |
| XLNet | 0.9852 | 0.9353 | 0.9107 | 0.9437 | 0.9852 | 0.9817 | 0.9880 | 0.9816 | 0.9841 | 0.9668 |
| DeBERTa | 0.9851 | 0.9837 | 0.9404 | 0.9697 | 0.9944 | 0.9890 | 0.9920 | 0.9820 | 0.9894 | 0.9810 |
| RoBERTa | 0.9880 | 0.9680 | 0.8335 | 0.9298 | 0.8292 | 0.7834 | 0.9675 | 0.9942 | 0.8936 | 0.9091 |
| DistilBERT | 0.9538 | 0.9445 | 0.9015 | 0.9333 | 0.9774 | 0.9856 | 0.9856 | 0.9899 | 0.9846 | 0.9626 |
| XLM-RoBERTa | 0.9659 | 0.9492 | 0.8789 | 0.9313 | 0.9760 | 0.9868 | 0.9861 | 0.9949 | 0.9860 | 0.9625 |
| Mistral | 0.9863 | 0.9841 | 0.8704 | 0.9469 | 0.9689 | 0.9322 | 0.9706 | 0.6207 | 0.8731 | 0.9047 |
| Llama | 0.9618 | 0.9814 | 0.9015 | 0.9482 | 0.9237 | 0.9552 | 0.9905 | 0.8484 | 0.9295 | 0.9375 |
| ImBD | 0.8726 | 0.7924 | 0.8094 | 0.8248 | 0.9526 | 0.8416 | 0.9582 | 0.8271 | 0.8949 | 0.8648 |
| DPIC | 0.9088 | 0.8320 | 0.8066 | 0.8491 | 0.8711 | 0.9034 | 0.9146 | 0.8654 | 0.8886 | 0.8717 |
| DetectAnyLLM | 0.9609 | 0.8097 | 0.6301 | 0.8002 | 0.5966 | 0.7683 | 0.9453 | 0.9058 | 0.8040 | 0.8024 |
| MPU | 0.5147 | 0.5084 | 0.4218 | 0.4816 | 0.4993 | 0.4975 | 0.4958 | 0.5061 | 0.4997 | 0.4919 |

Table C.4: The detection performance (AUPR) of various detectors in Cross-Model (CM) experiment.

| AUPR | Llama3.1-8b | DeepSeek-chat | gpt4o-mini | Qwen-plus | Avg. |
|---|---|---|---|---|---|
| *Zero-shot Methods* | | | | | |
| Likelihood | 0.4233 | 0.5629 | 0.5281 | 0.5574 | 0.5179 |
| Rank | 0.5030 | 0.5538 | 0.5443 | 0.5573 | 0.5396 |
| Logrank | 0.4569 | 0.5627 | 0.5287 | 0.5582 | 0.5266 |
| Entropy | 0.5040 | 0.4417 | 0.4422 | 0.4299 | 0.4545 |
| DetectGPT | 0.4152 | 0.5175 | 0.5695 | 0.5671 | 0.5173 |
| Fast-DetectGPT | 0.5343 | 0.6739 | 0.6956 | 0.6300 | 0.6335 |
| *Supervised Methods* | | | | | |
| GLTR-based | 0.7087 | 0.7510 | 0.7484 | 0.7453 | 0.7384 |
| PPL-based | 0.7661 | 0.8009 | 0.8197 | 0.8002 | 0.7967 |
| Ghostbuster | 0.7001 | 0.8188 | 0.7356 | 0.8083 | 0.7657 |
| Likelihood-based | 0.3591 | 0.7500 | 0.3559 | 0.3359 | 0.4502 |
| Rank-based | 0.6811 | 0.6517 | 0.6689 | 0.6699 | 0.6679 |
| Logrank-based | 0.7421 | 0.7164 | 0.7185 | 0.7265 | 0.7259 |
| Entropy-based | 0.6202 | 0.6396 | 0.6181 | 0.6282 | 0.6265 |
| XLNet | 0.9523 | 0.9952 | 0.9933 | 0.9876 | 0.9821 |
| DeBERTa | 0.9510 | 0.9983 | 0.9981 | 0.9914 | 0.9847 |
| RoBERTa | 0.9479 | 0.9967 | 0.9972 | 0.9916 | 0.9834 |
| DistilBERT | 0.9578 | 0.9903 | 0.9878 | 0.9793 | 0.9788 |
| XLM-RoBERTa | 0.9549 | 0.9946 | 0.9918 | 0.9683 | 0.9774 |
| Mistral | 0.9675 | 0.9792 | 0.9877 | 0.9749 | 0.9773 |
| Llama | 0.9474 | 0.9876 | 0.9862 | 0.9745 | 0.9739 |
| ImBD | 0.7799 | 0.9414 | 0.9402 | 0.9322 | 0.8984 |
| DPIC | 0.9134 | 0.9513 | 0.9253 | 0.9392 | 0.9323 |
| DetectAnyLLM | 0.9364 | 0.9730 | 0.9784 | 0.9725 | 0.9651 |
| MPU | 0.5480 | 0.4534 | 0.6040 | 0.4898 | 0.5238 |

Table C.5: The detection performance (AUPR) of various detectors in Cross-Operation (CO) experiment, grouped by operation types.

| AUPR | Creative Reformation | | | | Linguistic Enhancement | | | | Cross-lingual | Overall Avg. |
|---|---|---|---|---|---|---|---|---|---|---|
| | create | rewrite | summary | Avg. | polish | refine | expand | Avg. | translate | |
| *Zero-shot Methods* | | | | | | | | | | |
| Likelihood | 0.7170 | 0.4817 | 0.3910 | 0.5299 | 0.5251 | 0.4446 | 0.5798 | 0.5165 | 0.4826 | 0.5174 |
| Rank | 0.6174 | 0.5341 | 0.4290 | 0.5268 | 0.5379 | 0.4764 | 0.6493 | 0.5545 | 0.5309 | 0.5393 |
| Logrank | 0.7546 | 0.4814 | 0.3868 | 0.5409 | 0.5250 | 0.4411 | 0.5966 | 0.5209 | 0.4859 | 0.5245 |
| Entropy | 0.3824 | 0.4459 | 0.5566 | 0.4616 | 0.4430 | 0.4900 | 0.3944 | 0.4425 | 0.4818 | 0.4563 |
| DetectGPT | 0.6452 | 0.5725 | 0.4239 | 0.5472 | 0.5402 | 0.5186 | 0.6481 | 0.5690 | 0.5083 | 0.5510 |
| Fast-DetectGPT | 0.9135 | 0.5356 | 0.3950 | 0.6147 | 0.6116 | 0.4225 | 0.7557 | 0.5966 | 0.5876 | 0.6031 |
| *Supervised Methods* | | | | | | | | | | |
| GLTR-based | 0.6733 | 0.6865 | 0.6833 | 0.6810 | 0.6823 | 0.5978 | 0.7257 | 0.6686 | 0.6502 | 0.6713 |
| PPL-based | 0.7661 | 0.7806 | 0.8197 | 0.7983 | 0.7657 | 0.7443 | 0.8056 | 0.7719 | 0.7311 | 0.7774 |
| Ghostbuster | 0.7066 | 0.7528 | 0.5677 | 0.6757 | 0.7093 | 0.6535 | 0.7203 | 0.6944 | 0.7002 | 0.6872 |
| Likelihood-based | 0.2901 | 0.2833 | 0.4819 | 0.3518 | 0.2731 | 0.7495 | 0.7500 | 0.5909 | 0.2499 | 0.4397 |
| Rank-based | 0.5355 | 0.6278 | 0.6732 | 0.6122 | 0.6182 | 0.6183 | 0.5623 | 0.5996 | 0.6231 | 0.6083 |
| Logrank-based | 0.5531 | 0.6969 | 0.8023 | 0.6841 | 0.6903 | 0.7153 | 0.5712 | 0.6589 | 0.7067 | 0.6765 |
| Entropy-based | 0.5880 | 0.6087 | 0.6306 | 0.6091 | 0.6189 | 0.6048 | 0.5709 | 0.5982 | 0.5916 | 0.6019 |
| XLNet | 0.9953 | 0.9945 | 0.9932 | 0.9943 | 0.9943 | 0.9914 | 0.9892 | 0.9916 | 0.8639 | 0.9745 |
| DeBERTa | 0.9992 | 0.9995 | 0.9968 | 0.9985 | 0.9977 | 0.9971 | 0.9970 | 0.9973 | 0.9400 | 0.9896 |
| RoBERTa | 0.9981 | 0.9981 | 0.9959 | 0.9974 | 0.9988 | 0.9984 | 0.9961 | 0.9978 | 0.9263 | 0.9874 |
| DistilBERT | 0.9967 | 0.9954 | 0.9847 | 0.9923 | 0.9806 | 0.9855 | 0.9846 | 0.9836 | 0.8481 | 0.9679 |
| XLM-RoBERTa | 0.9975 | 0.9963 | 0.9929 | 0.9956 | 0.9919 | 0.9943 | 0.9942 | 0.9935 | 0.8943 | 0.9802 |
| Mistral | 0.9328 | 0.9266 | 0.9209 | 0.9268 | 0.9737 | 0.9854 | 0.9872 | 0.9821 | 0.8856 | 0.9446 |
| Llama | 0.9959 | 0.9935 | 0.9932 | 0.9942 | 0.9860 | 0.9864 | 0.9923 | 0.9882 | 0.8662 | 0.9734 |
| ImBD | 0.9940 | 0.9706 | 0.9192 | 0.9613 | 0.9291 | 0.8636 | 0.9136 | 0.9021 | 0.7806 | 0.9101 |
| DPIC | 0.9202 | 0.9542 | 0.9324 | 0.9356 | 0.9020 | 0.9366 | 0.9398 | 0.9261 | 0.7754 | 0.9087 |
| DetectAnyLLM | 0.9876 | 0.9861 | 0.9187 | 0.9641 | 0.9873 | 0.9720 | 0.9919 | 0.9837 | 0.8768 | 0.9601 |
| MPU | 0.5433 | 0.4991 | 0.4405 | 0.4943 | 0.5828 | 0.5107 | 0.5056 | 0.5330 | 0.4987 | 0.5115 |

Table C.6: The detection performance (AUC) of various detectors in Multi-Length (ML) experiment, grouped by formality levels (Formal vs. Informal) with overall average.

| | AUC | Formal | | | Informal | | | |
|---|---|---|---|---|---|---|---|---|
| Length | Detectors | xsum | pubmed | squad | wp | review | blog | tweets |
| | Likelihood | 0.4804 | 0.4418 | 0.4814 | 0.5778 | 0.5293 | 0.6110 | 0.8266 |
| | Rank | 0.4763 | 0.4597 | 0.4687 | 0.5380 | 0.5123 | 0.5860 | 0.8024 |
| | Logrank | 0.4873 | 0.4535 | 0.4851 | 0.5670 | 0.5158 | 0.6007 | 0.8104 |
| | Entropy | 0.4951 | 0.5466 | 0.4843 | 0.4825 | 0.4507 | 0.4671 | 0.2652 |
| | DetectGPT | 0.5000 | 0.5000 | 0.5000 | 0.5000 | 0.5000 | 0.5000 | 0.5000 |
| | Fast-DetectGPT | 0.5065 | 0.4582 | 0.4699 | 0.5746 | 0.4548 | 0.6045 | 0.7005 |
| | GLTR-based | 0.5009 | 0.5055 | 0.4752 | 0.4798 | 0.4427 | 0.5895 | 0.6134 |
| Len=10 | PPL-based | 0.5451 | 0.5839 | 0.5430 | 0.4377 | 0.5627 | 0.5239 | 0.6308 |
| | Ghostbuster | 0.5126 | 0.4979 | 0.4999 | 0.4818 | 0.4709 | 0.4777 | 0.4327 |
| | Likelihood-based | 0.5148 | 0.4936 | 0.4959 | 0.4431 | 0.5274 | 0.5063 | 0.7431 |
| | Rank-based | 0.4742 | 0.5028 | 0.4735 | 0.5039 | 0.4924 | 0.5584 | 0.5901 |
| | Logrank-based | 0.4915 | 0.5006 | 0.4722 | 0.4948 | 0.5128 | 0.5252 | 0.8266 |
| | Entropy-based | 0.5157 | 0.5065 | 0.5316 | 0.4142 | 0.5164 | 0.4711 | 0.4921 |
| | XLNet | 0.6152 | 0.5706 | 0.5561 | 0.5342 | 0.5563 | 0.6083 | 0.7016 |
| | DeBERTa | 0.6614 | 0.7076 | 0.6677 | 0.7058 | 0.6417 | 0.6726 | 0.9011 |
| | RoBERTa | 0.6662 | 0.6698 | 0.6442 | 0.6581 | 0.6446 | 0.7215 | 0.9509 |
| | DistilBERT | 0.6620 | 0.6735 | 0.6518 | 0.6678 | 0.6366 | 0.7061 | 0.9092 |
| | XLM-RoBERTa | 0.6392 | 0.6697 | 0.6247 | 0.6535 | 0.6407 | 0.6770 | 0.9443 |
| | Mistral | 0.5150 | 0.5025 | 0.4995 | 0.4321 | 0.5219 | 0.4993 | 0.4553 |
| | Llama | 0.5247 | 0.5122 | 0.5072 | 0.4468 | 0.5119 | 0.4949 | 0.5628 |
| | ImBD | 0.5878 | 0.5133 | 0.5649 | 0.6527 | 0.5238 | 0.7467 | 0.7293 |
| | DPIC | 0.5752 | 0.5670 | 0.5473 | 0.5761 | 0.5198 | 0.6459 | 0.7211 |
| | DetectAnyLLM | 0.5271 | 0.4751 | 0.5044 | 0.5705 | 0.4798 | 0.6540 | 0.8507 |
| | MPU | 0.5058 | 0.4939 | 0.4755 | 0.5019 | 0.5094 | 0.4898 | 0.5008 |
| | Likelihood | 0.4489 | 0.4654 | 0.4569 | 0.6675 | 0.5368 | 0.6660 | 0.8646 |
| | Rank | 0.4680 | 0.4885 | 0.4676 | 0.5925 | 0.5167 | 0.6170 | 0.8390 |
| | Logrank | 0.4554 | 0.4803 | 0.4635 | 0.6508 | 0.5203 | 0.6573 | 0.8531 |
| | Entropy | 0.5406 | 0.5831 | 0.5272 | 0.4211 | 0.4109 | 0.3926 | 0.1709 |
| | DetectGPT | 0.4785 | 0.5069 | 0.5243 | 0.6239 | 0.6196 | 0.5140 | 0.6673 |
| | Fast-DetectGPT | 0.4730 | 0.5300 | 0.4772 | 0.6404 | 0.4663 | 0.6164 | 0.7115 |
| | GLTR-based | 0.5795 | 0.5641 | 0.5163 | 0.4257 | 0.5368 | 0.5095 | 0.7084 |
| Len=50 | PPL-based | 0.6057 | 0.6742 | 0.6032 | 0.4630 | 0.4799 | 0.5282 | 0.7793 |
| | Ghostbuster | 0.5588 | 0.5763 | 0.6176 | 0.6484 | 0.5588 | 0.5175 | 0.6166 |
| | Likelihood-based | 0.5448 | 0.5738 | 0.5297 | 0.4409 | 0.5003 | 0.4036 | 0.6647 |
| | Rank-based | 0.5182 | 0.4811 | 0.5035 | 0.5227 | 0.5011 | 0.4773 | 0.6463 |
| | Logrank-based | 0.5342 | 0.5902 | 0.5341 | 0.4732 | 0.5124 | 0.4102 | 0.7912 |
| | Entropy-based | 0.5105 | 0.5359 | 0.5354 | 0.5549 | 0.4970 | 0.5106 | 0.6620 |
| | XLNet | 0.8937 | 0.8364 | 0.8333 | 0.8918 | 0.9090 | 0.9359 | 0.9759 |
| | DeBERTa | 0.9225 | 0.9148 | 0.8953 | 0.9527 | 0.9419 | 0.9274 | 0.9741 |
| | RoBERTa | 0.8951 | 0.8400 | 0.6952 | 0.8380 | 0.9101 | 0.8471 | 0.9965 |
| | DistilBERT | 0.8536 | 0.8458 | 0.8304 | 0.8647 | 0.9033 | 0.9116 | 0.9847 |
| | XLM-RoBERTa | 0.8086 | 0.8429 | 0.7721 | 0.8905 | 0.8759 | 0.8765 | 0.9914 |
| | Mistral | 0.8883 | 0.8786 | 0.7219 | 0.7351 | 0.7884 | 0.8145 | 0.3337 |
| | Llama | 0.8746 | 0.8638 | 0.8197 | 0.8095 | 0.8345 | 0.8825 | 0.7028 |
| | ImBD | 0.7715 | 0.6725 | 0.7142 | 0.8137 | 0.7352 | 0.8929 | 0.7946 |
| | DPIC | 0.7701 | 0.7231 | 0.7408 | 0.7476 | 0.7169 | 0.8143 | 0.8704 |
| | DetectAnyLLM | 0.7367 | 0.6610 | 0.5384 | 0.6739 | 0.6092 | 0.8580 | 0.8836 |
| | MPU | 0.5260 | 0.4949 | 0.4854 | 0.5033 | 0.5028 | 0.4978 | 0.5064 |

Table C.7: The detection performance (AUC) of various detectors in Multi-Length (ML) experiment, grouped by formality levels (Formal vs. Informal) with overall average.

| | AUC | Formal | | | Informal | | | |
|---|---|---|---|---|---|---|---|---|
| **Length** | **Detectors** | **xsum** | **pubmed** | **squad** | **wp** | **review** | **blog** | **tweets** |
| | Likelihood | 0.4197 | 0.4612 | 0.4591 | 0.6819 | 0.5651 | 0.6740 | 0.8671 |
| | Rank | 0.4497 | 0.4905 | 0.4647 | 0.6120 | 0.5313 | 0.6423 | 0.8453 |
| | Logrank | 0.4258 | 0.4742 | 0.4650 | 0.6678 | 0.5486 | 0.6672 | 0.8559 |
| | Entropy | 0.5656 | 0.6028 | 0.5244 | 0.3799 | 0.3705 | 0.3607 | 0.1660 |
| | DetectGPT | 0.4592 | 0.4643 | 0.5174 | 0.6542 | 0.5787 | 0.5187 | 0.6799 |
| | Fast-DetectGPT | 0.4225 | 0.5565 | 0.4833 | 0.6432 | 0.4853 | 0.6176 | 0.7148 |
| | GLTR-based | 0.5781 | 0.6198 | 0.5924 | 0.5271 | 0.4967 | 0.5552 | 0.7051 |
| Len=100 | PPL-based | 0.6498 | 0.6945 | 0.6552 | 0.5269 | 0.5767 | 0.6267 | 0.7730 |
| | Ghostbuster | 0.6216 | 0.5734 | 0.6752 | 0.7336 | 0.6642 | 0.6234 | 0.6283 |
| | Likelihood-based | 0.5367 | 0.5672 | 0.5453 | 0.4912 | 0.4955 | 0.3910 | 0.6565 |
| | Rank-based | 0.5087 | 0.4895 | 0.5008 | 0.5341 | 0.5020 | 0.4527 | 0.6454 |
| | Logrank-based | 0.5336 | 0.5951 | 0.5401 | 0.4699 | 0.5051 | 0.4171 | 0.7868 |
| | Entropy-based | 0.4997 | 0.5440 | 0.5352 | 0.5181 | 0.5223 | 0.4501 | 0.6621 |
| | XLNet | 0.9759 | 0.9081 | 0.8953 | 0.9644 | 0.9585 | 0.9744 | 0.9767 |
| | DeBERTa | 0.9607 | 0.9674 | 0.9296 | 0.9794 | 0.9716 | 0.9810 | 0.9777 |
| | RoBERTa | 0.9406 | 0.9084 | 0.7409 | 0.9250 | 0.9601 | 0.9424 | 0.9965 |
| | DistilBERT | 0.9094 | 0.9134 | 0.8811 | 0.9393 | 0.9584 | 0.9641 | 0.9851 |
| | XLM-RoBERTa | 0.9218 | 0.9251 | 0.8399 | 0.9548 | 0.9516 | 0.9637 | 0.9921 |
| | Mistral | 0.9596 | 0.9589 | 0.8020 | 0.9017 | 0.8518 | 0.8894 | 0.3371 |
| | Llama | 0.9336 | 0.9470 | 0.8731 | 0.9389 | 0.9097 | 0.9512 | 0.7185 |
| | ImBD | 0.8210 | 0.7242 | 0.7583 | 0.8618 | 0.8081 | 0.9318 | 0.7782 |
| | DPIC | 0.8529 | 0.7832 | 0.8071 | 0.8172 | 0.8259 | 0.8771 | 0.8714 |
| | DetectAnyLLM | 0.8697 | 0.7656 | 0.5275 | 0.6978 | 0.7415 | 0.9254 | 0.8838 |
| | MPU | 0.5124 | 0.5139 | 0.4513 | 0.5024 | 0.4900 | 0.4900 | 0.5069 |
| | Likelihood | 0.4131 | 0.4284 | 0.4737 | 0.6704 | 0.5675 | 0.6787 | 0.8671 |
| | Rank | 0.4271 | 0.4662 | 0.4799 | 0.6250 | 0.5455 | 0.6658 | 0.8454 |
| | Logrank | 0.4154 | 0.4414 | 0.4801 | 0.6577 | 0.5511 | 0.6706 | 0.8559 |
| | Entropy | 0.5703 | 0.6340 | 0.5079 | 0.3604 | 0.3694 | 0.3382 | 0.1659 |
| | DetectGPT | 0.4373 | 0.4644 | 0.5395 | 0.6684 | 0.5680 | 0.5704 | 0.6778 |
| | Fast-DetectGPT | 0.3995 | 0.5598 | 0.4937 | 0.6453 | 0.4845 | 0.6275 | 0.7147 |
| | GLTR-based | 0.5965 | 0.5560 | 0.5943 | 0.6820 | 0.6542 | 0.6515 | 0.7049 |
| Len=200 | PPL-based | 0.6559 | 0.6165 | 0.6644 | 0.5964 | 0.6914 | 0.6855 | 0.7726 |
| | Ghostbuster | 0.6090 | 0.5924 | 0.6609 | 0.7999 | 0.7380 | 0.6535 | 0.6294 |
| | Likelihood-based | 0.5330 | 0.5133 | 0.5420 | 0.5780 | 0.5243 | 0.4454 | 0.6569 |
| | Rank-based | 0.4986 | 0.5383 | 0.4953 | 0.5785 | 0.5162 | 0.4876 | 0.6455 |
| | Logrank-based | 0.5286 | 0.5424 | 0.5414 | 0.5351 | 0.5204 | 0.4574 | 0.7871 |
| | Entropy-based | 0.4865 | 0.5183 | 0.5117 | 0.5632 | 0.5496 | 0.4576 | 0.6612 |
| | XLNet | 0.9650 | 0.9264 | 0.9024 | 0.9827 | 0.9778 | 0.9828 | 0.9767 |
| | DeBERTa | 0.9757 | 0.9810 | 0.9376 | 0.9909 | 0.9854 | 0.9890 | 0.9777 |
| | RoBERTa | 0.9563 | 0.9422 | 0.7605 | 0.9647 | 0.9758 | 0.9755 | 0.9965 |
| | DistilBERT | 0.9374 | 0.9352 | 0.8923 | 0.9694 | 0.9804 | 0.9780 | 0.9851 |
| | XLM-RoBERTa | 0.9505 | 0.9468 | 0.8570 | 0.9713 | 0.9799 | 0.9823 | 0.9923 |
| | Mistral | 0.9739 | 0.9779 | 0.8356 | 0.9605 | 0.9094 | 0.9352 | 0.3373 |
| | Llama | 0.9572 | 0.9646 | 0.8878 | 0.9676 | 0.9552 | 0.9722 | 0.7189 |
| | ImBD | 0.8465 | 0.7421 | 0.7734 | 0.8973 | 0.8390 | 0.9492 | 0.7777 |
| | DPIC | 0.8980 | 0.8082 | 0.8235 | 0.8599 | 0.8854 | 0.9107 | 0.8711 |
| | DetectAnyLLM | 0.9099 | 0.8000 | 0.5479 | 0.6909 | 0.8293 | 0.9471 | 0.8838 |
| | MPU | 0.5083 | 0.4948 | 0.3870 | 0.5024 | 0.4992 | 0.4921 | 0.5066 |

Table C.8: The detection performance (AUC) of various detectors in Multi-Length (ML) experiment, grouped by formality levels (Formal vs. Informal) with overall average.

| AUC | | Formal | | | Informal | | | |
| | Detectors | xsum | pubmed | squad | wp | review | blog | tweets |
|---|---|---|---|---|---|---|---|---|
| Length | Likelihood | 0.4162 | 0.4378 | 0.4847 | 0.6328 | 0.5617 | 0.6769 | 0.8671 |
| | Rank | 0.4192 | 0.4816 | 0.4905 | 0.6187 | 0.5667 | 0.6788 | 0.8454 |
| | Logrank | 0.4152 | 0.4533 | 0.4919 | 0.6224 | 0.5476 | 0.6682 | 0.8559 |
| | Entropy | 0.5659 | 0.6237 | 0.4973 | 0.3741 | 0.3895 | 0.3302 | 0.1659 |
| | DetectGPT | 0.4717 | 0.4772 | 0.5393 | 0.6244 | 0.5849 | 0.5567 | 0.6778 |
| | Fast-DetectGPT | 0.3967 | 0.5652 | 0.5061 | 0.6162 | 0.4859 | 0.6296 | 0.7147 |
| | GLTR-based | 0.5421 | 0.5536 | 0.6031 | 0.6880 | 0.6749 | 0.6778 | 0.7049 |
| Len=500 | PPL-based | 0.5977 | 0.6110 | 0.6645 | 0.6243 | 0.7229 | 0.7174 | 0.7726 |
| | Ghostbuster | 0.6433 | 0.5917 | 0.6476 | 0.7763 | 0.7419 | 0.6939 | 0.6294 |
| | Likelihood-based | 0.4757 | 0.5148 | 0.5421 | 0.6501 | 0.5291 | 0.4835 | 0.6569 |
| | Rank-based | 0.5077 | 0.5600 | 0.4961 | 0.6376 | 0.5252 | 0.5165 | 0.6455 |
| | Logrank-based | 0.4926 | 0.5412 | 0.5392 | 0.6072 | 0.5315 | 0.4999 | 0.7871 |
| | Entropy-based | 0.4605 | 0.5377 | 0.5137 | 0.5864 | 0.5668 | 0.4892 | 0.6612 |
| | XLNet | 0.9699 | 0.9290 | 0.9025 | 0.9852 | 0.9802 | 0.9862 | 0.9767 |
| | DeBERTa | 0.9796 | 0.9821 | 0.9385 | 0.9938 | 0.9880 | 0.9911 | 0.9777 |
| | RoBERTa | 0.9602 | 0.9474 | 0.7613 | 0.9740 | 0.9771 | 0.9782 | 0.9965 |
| | DistilBERT | 0.9433 | 0.9380 | 0.8934 | 0.9765 | 0.9841 | 0.9827 | 0.9851 |
| | XLM-RoBERTa | 0.9577 | 0.9471 | 0.8580 | 0.9750 | 0.9859 | 0.9851 | 0.9923 |
| | Mistral | 0.9783 | 0.9773 | 0.8377 | 0.9737 | 0.9327 | 0.9496 | 0.3373 |
| | Llama | 0.9625 | 0.9665 | 0.8881 | 0.9762 | 0.9632 | 0.9769 | 0.7189 |
| | ImBD | 0.8321 | 0.7473 | 0.7793 | 0.8976 | 0.8210 | 0.9487 | 0.7777 |
| | DPIC | 0.9091 | 0.8240 | 0.8253 | 0.8796 | 0.8963 | 0.9233 | 0.8711 |
| | DetectAnyLLM | 0.9201 | 0.8143 | 0.5493 | 0.6131 | 0.8498 | 0.9499 | 0.8838 |
| | MPU | 0.5073 | 0.5116 | 0.3852 | 0.5005 | 0.4978 | 0.5008 | 0.5066 |

Table C.9: The detection performance (AUC) of various detectors in LLM-Va: small v.s. large experiments.

| AUC | Small | | | | Large | | |
| | gpt4o-mini | Qwen2-7b-instruct | Qwen3-8b | Avg. | gpt4o | Qwen-plus | Avg. |
|---|---|---|---|---|---|---|---|
| Likelihood | 0.5679 | 0.5152 | 0.5685 | 0.5505 | 0.5578 | 0.5854 | 0.5716 |
| Rank | 0.5786 | 0.5374 | 0.5902 | 0.5687 | 0.5808 | 0.5844 | 0.5826 |
| Logrank | 0.5642 | 0.5023 | 0.5676 | 0.5447 | 0.5500 | 0.5787 | 0.5643 |
| Entropy | 0.4451 | 0.4952 | 0.4603 | 0.4668 | 0.4547 | 0.4152 | 0.4349 |
| DetectGPT | 0.5100 | 0.5342 | 0.5071 | 0.5171 | 0.5160 | 0.5114 | 0.5137 |
| Fast-DetectGPT | 0.6630 | 0.5646 | 0.6910 | 0.6395 | 0.6354 | 0.5996 | 0.6175 |
| GLTR-based | 0.7368 | 0.7371 | 0.7265 | 0.7334 | 0.7250 | 0.7363 | 0.7306 |
| PPL-based | 0.8184 | 0.8375 | 0.7802 | 0.8120 | 0.8037 | 0.7922 | 0.7979 |
| Ghostbuster | 0.7923 | 0.7994 | 0.7389 | 0.7768 | 0.7923 | 0.8045 | 0.7984 |
| Likelihood-based | 0.6727 | 0.6804 | 0.6426 | 0.6652 | 0.6760 | 0.6713 | 0.6736 |
| Rank-based | 0.6808 | 0.6958 | 0.6756 | 0.6840 | 0.6738 | 0.6918 | 0.6828 |
| Logrank-based | 0.7326 | 0.7463 | 0.7037 | 0.7275 | 0.7373 | 0.7373 | 0.7373 |
| Entropy-based | 0.6227 | 0.6112 | 0.6012 | 0.6117 | 0.6099 | 0.6108 | 0.6103 |
| XLNet | 0.9942 | 0.9916 | 0.9772 | 0.9876 | 0.9898 | 0.9849 | 0.9873 |
| DeBERTa | 0.9972 | 0.9950 | 0.9857 | 0.9926 | 0.9955 | 0.9882 | 0.9918 |
| RoBERTa | 0.9953 | 0.9939 | 0.9887 | 0.9926 | 0.9941 | 0.9864 | 0.9902 |
| DistilBERT | 0.9828 | 0.9793 | 0.9472 | 0.9697 | 0.9734 | 0.9735 | 0.9734 |
| XLM-RoBERTa | 0.9894 | 0.9873 | 0.9756 | 0.9841 | 0.9851 | 0.9815 | 0.9833 |
| Mistral | 0.9163 | 0.9881 | 0.9591 | 0.9545 | 0.9087 | 0.9707 | 0.9397 |
| Llama | 0.9761 | 0.9811 | 0.9483 | 0.9685 | 0.9660 | 0.9624 | 0.9642 |
| ImBD | 0.9364 | 0.9318 | 0.8839 | 0.9174 | 0.9351 | 0.9350 | 0.9350 |
| DPIC | 0.9162 | 0.6404 | 0.5799 | 0.7122 | 0.9112 | 0.6134 | 0.7623 |
| DetectAnyLLM | 0.9819 | 0.9737 | 0.9609 | 0.9721 | 0.9747 | 0.9670 | 0.9708 |
| MPU | 0.6085 | 0.4554 | 0.4571 | 0.5070 | 0.9900 | 0.4763 | 0.7331 |
| Overall Avg. | 0.7783 | 0.7572 | 0.7465 | 0.7607 | 0.7890 | 0.7565 | 0.7728 |

## C.6 EVALUATION PERFORMANCE (AUC) UNDER DIFFERENT TRAINING CONDITIONS

To investigate embedding-based detectors, while further discover how to effectively train a supervised detector, we evaluated their performance under varied training conditions. We trained XLNet, DeBERTa, RoBERTa, distilBert, xlm-roberta, Mistral and Llama under the condition of different training epochs and loss thresholds, with detailed results shows below in Table C.17, C.18,and C.19.

Table C.10: The detection performance (AUC) of various detectors in LLM-Va: instruct v.s. base experiments.

| AUC | Llama | | | Qwen | | |
|---|---|---|---|---|---|---|
| | Llama3.1-8b | Llama3.1-8b-instruct | Avg. | Qwen2-7b-instruct | Qwen3-8b | Avg. |
| Likelihood | 0.3990 | 0.3380 | 0.3685 | 0.5152 | 0.5685 | 0.5418 |
| Rank | 0.5089 | 0.4929 | 0.5009 | 0.5374 | 0.5902 | 0.5638 |
| Logrank | 0.4417 | 0.3798 | 0.4107 | 0.5023 | 0.5676 | 0.5349 |
| Entropy | 0.5553 | 0.5998 | 0.5775 | 0.4952 | 0.4603 | 0.4777 |
| DetectGPT | 0.4328 | 0.4049 | 0.4188 | 0.5342 | 0.5071 | 0.5206 |
| Fast-DetectGPT | 0.3936 | 0.3104 | 0.3520 | 0.5646 | 0.6910 | 0.6278 |
| GLTR-based | 0.6969 | 0.6745 | 0.6857 | 0.7371 | 0.7265 | 0.7318 |
| PPL-based | 0.7533 | 0.7260 | 0.7397 | 0.8375 | 0.7802 | 0.8088 |
| Ghostbuster | 0.7435 | 0.7222 | 0.7328 | 0.7994 | 0.7389 | 0.7691 |
| Likelihood-based | 0.6742 | 0.6498 | 0.6620 | 0.6804 | 0.6426 | 0.6615 |
| Rank-based | 0.6997 | 0.6735 | 0.6866 | 0.6958 | 0.6756 | 0.6857 |
| Logrank-based | 0.7504 | 0.7520 | 0.7512 | 0.7463 | 0.7037 | 0.7250 |
| Entropy-based | 0.6227 | 0.6112 | 0.5976 | 0.6117 | 0.6012 | 0.6062 |
| XLNet | 0.9330 | 0.8908 | 0.9119 | 0.9916 | 0.9772 | 0.9844 |
| DeBERTa | 0.9576 | 0.9028 | 0.9302 | 0.9950 | 0.9857 | 0.9903 |
| RoBERTa | 0.9025 | 0.8756 | 0.8890 | 0.9939 | 0.9887 | 0.9913 |
| DistilBERT | 0.9422 | 0.9042 | 0.9232 | 0.9793 | 0.9472 | 0.9632 |
| XLM-RoBERTa | 0.9442 | 0.9061 | 0.9252 | 0.9873 | 0.9756 | 0.9814 |
| Mistral | 0.9575 | 0.9359 | 0.9467 | 0.9881 | 0.9591 | 0.9736 |
| Llama | 0.9509 | 0.9278 | 0.9393 | 0.9811 | 0.9483 | 0.9647 |
| ImBD | 0.7990 | 0.7570 | 0.7780 | 0.9318 | 0.8839 | 0.9079 |
| DPIC | 0.9104 | 0.8908 | 0.9006 | 0.6404 | 0.5799 | 0.6102 |
| DetectAnyLLM | 0.9165 | 0.8736 | 0.8950 | 0.9737 | 0.9609 | 0.9673 |
| MPU | 0.5861 | 0.5671 | 0.5766 | 0.4554 | 0.4571 | 0.4562 |
| Overall Avg. | 0.7272 | 0.6977 | 0.7125 | 0.7572 | 0.7465 | 0.7519 |

Table C.11: The detection performance (AUC) of various detectors in LLM-Va: reasoning v.s. non-reasoning experiments.

| AUC | GPT | | | Qwen | | Deepseek | |
|---|---|---|---|---|---|---|---|
| | gpt4o-mini | gpt5 | gpt5-rs | Qwen-plus | Qwen-plus-rs | DeepSeek-chat | DeepSeek-rs |
| Likelihood | 0.5679 | 0.4821 | 0.4893 | 0.5854 | 0.5289 | 0.5744 | 0.5252 |
| Rank | 0.5786 | 0.5530 | 0.5441 | 0.5844 | 0.5590 | 0.5782 | 0.5596 |
| Logrank | 0.5642 | 0.4749 | 0.4815 | 0.5787 | 0.5122 | 0.5633 | 0.5159 |
| Entropy | 0.4451 | 0.5238 | 0.5054 | 0.4152 | 0.4787 | 0.4377 | 0.4867 |
| DetectGPT | 0.5100 | 0.5362 | 0.5323 | 0.5114 | 0.5884 | 0.4846 | 0.5810 |
| Fast-DetectGPT | 0.6630 | 0.4929 | 0.4627 | 0.5996 | 0.5201 | 0.6143 | 0.5968 |
| GLTR-based | 0.7408 | 0.6988 | 0.6670 | 0.7339 | 0.6930 | 0.7362 | 0.7315 |
| PPL-based | 0.8154 | 0.7382 | 0.7098 | 0.7846 | 0.7497 | 0.7784 | 0.7661 |
| Ghostbuster | 0.7827 | 0.6891 | 0.7014 | 0.8035 | 0.8346 | 0.8451 | 0.7928 |
| Likelihood-based | 0.6830 | 0.6804 | 0.6668 | 0.6809 | 0.6593 | 0.6361 | 0.6583 |
| Rank-based | 0.6905 | 0.6702 | 0.6549 | 0.6767 | 0.6576 | 0.6703 | 0.6624 |
| Logrank-based | 0.7318 | 0.7229 | 0.7171 | 0.7224 | 0.7177 | 0.6987 | 0.7066 |
| Entropy-based | 0.6185 | 0.5811 | 0.6053 | 0.6091 | 0.6024 | 0.6127 | 0.6159 |
| XLNet | 0.9934 | 0.9784 | 0.9730 | 0.9866 | 0.9936 | 0.9955 | 0.9936 |
| DeBERTa | 0.9976 | 0.9780 | 0.9707 | 0.9906 | 0.9978 | 0.9974 | 0.9967 |
| RoBERTa | 0.9965 | 0.9853 | 0.9841 | 0.9904 | 0.9976 | 0.9962 | 0.9956 |
| DistilBERT | 0.9856 | 0.9599 | 0.9506 | 0.9739 | 0.9840 | 0.9875 | 0.9824 |
| XLM-RoBERTa | 0.9892 | 0.9732 | 0.9693 | 0.9805 | 0.9927 | 0.9917 | 0.9896 |
| Mistral | 0.9830 | 0.9236 | 0.9072 | 0.9770 | 0.9817 | 0.9816 | 0.9694 |
| Llama | 0.9817 | 0.9247 | 0.9122 | 0.9699 | 0.9779 | 0.9810 | 0.9709 |
| ImBD | 0.9494 | 0.9104 | 0.8931 | 0.9460 | 0.9477 | 0.9461 | 0.9457 |
| DPIC | 0.9294 | 0.8993 | 0.8657 | 0.7063 | 0.7828 | 0.5059 | 0.5070 |
| DetectAnyLLM | 0.9852 | 0.9614 | 0.9631 | 0.9675 | 0.9758 | 0.9780 | 0.9728 |
| MPU | 0.4761 | 0.4397 | 0.4641 | 0.4527 | 0.4690 | 0.5331 | 0.5186 |
| Overall Avg. | 0.7774 | 0.7407 | 0.7329 | 0.7594 | 0.7584 | 0.7551 | 0.7517 |

We also trained supervised detectors on different scales of training data by reducing training set sizes to 2,000, 1,000, and 500 samples in order to discover the impact of training scales on the performance of detectors. The results are shown in Table C.20, C.21, C.22, C.23, C.24, and C.25

Table C.12: The detection performance (AUC) of various detectors in LLM-Co experiments.

| AUC | Llama3.1-8b | | | Deepseek-chat | | |
|---|---|---|---|---|---|---|
| | DeepSeek-chat | gpt4o-mini | Qwen-plus | Llama3.1-8b | gpt4o-mini | Qwen-plus |
| *Zero-shot Methods* | | | | | | |
| Likelihood | 0.1269 | 0.1385 | -0.0665 | 0.0059 | -0.0572 | -0.1036 |
| Rank | 0.0966 | 0.1102 | 0.1223 | 0.0615 | -0.0140 | 0.0263 |
| Logrank | 0.1030 | 0.1034 | 0.1252 | -0.0007 | -0.0684 | -0.0015 |
| Entropy | -0.0793 | -0.0820 | -0.0990 | -0.0422 | 0.0459 | 0.0034 |
| DetectGPT | 0.0256 | 0.0125 | -0.0487 | 0.0344 | 0.0198 | -0.0128 |
| Fast-DetectGPT | -0.0337 | 0.1932 | -0.1523 | -0.0411 | -0.0930 | -0.1186 |
| Zero-shot Avg. | 0.1005 | 0.1113 | 0.1214 | -0.0034 | -0.0278 | -0.0499 |
| *Supervised Methods* | | | | | | |
| GLTR-based | -0.0054 | -0.0101 | -0.0108 | 0.0182 | -0.0365 | -0.0502 |
| PPL-based | -0.0250 | -0.0322 | -0.0321 | 0.0057 | -0.0626 | -0.0478 |
| Ghostbuster | -0.0745 | -0.1200 | -0.0582 | 0.0712 | -0.0374 | -0.0475 |
| Likelihood-based | -0.1280 | -0.1208 | -0.1346 | -0.0052 | 0.0037 | 0.0117 |
| Rank-based | -0.0263 | -0.0339 | -0.0719 | -0.0085 | 0.0217 | -0.0075 |
| Logrank-based | -0.0740 | -0.0918 | -0.1380 | -0.0238 | -0.0708 | -0.0254 |
| Entropy-based | -0.0758 | -0.0521 | -0.0798 | -0.0815 | -0.0078 | -0.0300 |
| XLNet | -0.0270 | 0.0125 | 0.0005 | 0.0385 | 0.0001 | -0.0012 |
| DeBERTa | -0.0295 | -0.0034 | -0.0184 | -0.0116 | -0.0007 | -0.0026 |
| RoBERTa | -0.0071 | -0.0096 | -0.0054 | -0.0011 | -0.0024 | -0.0018 |
| DistilBERT | 0.0254 | 0.0213 | 0.0195 | 0.0179 | -0.0049 | -0.0052 |
| XLM-RoBERTa | -0.0261 | -0.0163 | -0.0097 | 0.0088 | -0.0024 | -0.0008 |
| Mistral | -0.0237 | 0.0374 | -0.0331 | -0.0474 | 0.0310 | 0.0136 |
| Llama | -0.0578 | -0.0261 | -0.0501 | 0.0342 | -0.0175 | -0.0412 |
| ImBD | 0.1380 | 0.1715 | 0.1736 | -0.2608 | 0.0154 | 0.0108 |
| DPIC | -0.1216 | -0.0733 | -0.1075 | -0.1360 | -0.0454 | -0.0195 |
| DetectAnyLLM | -0.0334 | 0.0095 | -0.0199 | -0.0926 | 0.0107 | 0.0100 |
| MPU | 0.0595 | 0.0624 | 0.0740 | 0.0361 | 0.0108 | -0.0120 |
| Supervised Avg. | -0.0297 | -0.0162 | -0.0289 | -0.1172 | 0.0005 | -0.0042 |
| Overall Avg. | 0.0029 | 0.0157 | 0.0087 | -0.1375 | -0.0007 | -0.0028 |

## C.7 EVALUATION PERFORMANCE (AUC) OF OPEN-ACCESS DETECTORS

We evaluated three open-access detectors to assess their generalization ability and detection performance, including RADAR, OpenAI's RoBERTa-based detector, RoBERTa-HC3, MPU, and ArguGPT. Results are shown in Table C.26, C.27 and C.28.

Table C.13: The detection performance (AUC) of various detectors in LLM-Co experiments.

| AUC | gpt4o-mini | | | Qwen-plus | | |
|---|---|---|---|---|---|---|
| | Llama3.1-8b | DeepSeek-chat | Qwen-plus | Llama3.1-8b | DeepSeek-chat | gpt4o-mini |
| *Zero-shot Methods* | | | | | | |
| Likelihood | 0.1269 | 0.1385 | 0.1521 | -0.2887 | -0.0091 | -0.0058 |
| Rank | 0.0966 | 0.1102 | 0.1223 | -0.1551 | 0.0222 | 0.0263 |
| Logrank | 0.1030 | 0.1034 | 0.1252 | -0.2441 | -0.0089 | -0.0015 |
| Entropy | -0.0793 | -0.0820 | -0.0990 | 0.2007 | 0.0126 | 0.0034 |
| DetectGPT | 0.0256 | 0.0125 | -0.0487 | 0.0344 | -0.0005 | 0.0131 |
| Fast-DetectGPT | -0.0337 | 0.1932 | -0.1523 | -0.4113 | 0.0396 | 0.0581 |
| Zero-shot Aavg. | -0.1744 | 0.0197 | 0.0320 | -0.1969 | 0.0120 | 0.0152 |
| *Supervised Methods* | | | | | | |
| GLTR-based | -0.0770 | 0.0196 | 0.0136 | -0.0866 | 0.0101 | 0.0121 |
| PPL-based | -0.0520 | -0.0322 | -0.0321 | -0.1711 | 0.0207 | 0.0278 |
| Ghostbuster | -0.0745 | -0.1200 | -0.0582 | -0.1596 | -0.0367 | -0.0097 |
| Likelihood-based | 0.0736 | -0.0212 | 0.0126 | -0.0735 | 0.0121 | 0.0064 |
| Rank-based | 0.0296 | 0.0057 | -0.0273 | 0.0445 | 0.0103 | 0.0219 |
| Logrank-based | 0.0515 | -0.0330 | -0.0080 | 0.0085 | -0.0030 | -0.0016 |
| Entropy-based | -0.0210 | -0.0176 | 0.0442 | -0.0187 | 0.0089 | -0.0103 |
| XLNet | -0.1230 | -0.0002 | 0.0186 | 0.0019 | 0.0094 | -0.0318 |
| DeBERTa | -0.0352 | -0.0334 | -0.0184 | -0.0116 | -0.0007 | -0.0026 |
| RoBERTa | -0.0071 | -0.0096 | -0.0054 | -0.2859 | -0.0003 | -0.0002 |
| DistilBERT | 0.0254 | 0.0213 | 0.0195 | -0.1591 | -0.0049 | -0.0052 |
| XLM-RoBERTa | -0.0261 | -0.0163 | -0.0097 | -0.2334 | 0.0167 | 0.0122 |
| Mistral | -0.0237 | 0.0374 | -0.0331 | -0.0474 | 0.0310 | 0.0136 |
| Llama | -0.0578 | -0.0261 | -0.0501 | 0.0028 | -0.0039 | 0.0049 |
| ImBD | -0.2204 | 0.0349 | 0.0450 | -0.3089 | 0.0390 | 0.0321 |
| DPIC | -0.0738 | -0.0032 | 0.0220 | -0.0496 | -0.0012 | -0.0036 |
| DetectAnyLLM | -0.0221 | -0.0300 | 0.0054 | -0.0268 | 0.0136 | 0.0210 |
| MPU | 0.0674 | -0.0462 | -0.0300 | -0.0551 | 0.0643 | 0.0317 |
| Supervised Avg. | -0.0745 | -0.0046 | 0.0093 | -0.0913 | 0.0186 | 0.0116 |
| Overall Avg. | -0.0994 | 0.0015 | 0.0150 | -0.1177 | 0.0169 | 0.0125 |

Table C.14: The detection performance (AUC) of various detectors in Op-Co experiments.

| AUC | create | | | | rewrite | | |
|---|---|---|---|---|---|---|---|
| | **polish** | **expand** | **refine** | **rewrite** | **polish** | **expand** | **refine** |
| *Zero-shot Methods* | | | | | | | |
| Likelihood | -0.1102 | -0.0265 | -0.2307 | -0.1549 | -0.0052 | 0.1060 | -0.0480 |
| Rank | 0.1014 | 0.1962 | 0.0105 | 0.0695 | -0.0098 | 0.1417 | -0.0537 |
| Logrank | -0.1199 | -0.0239 | -0.2460 | -0.1678 | -0.0067 | 0.1188 | -0.0540 |
| Entropy | 0.0827 | 0.0029 | 0.1886 | 0.1136 | 0.0093 | -0.0667 | 0.0500 |
| DetectGPT | -0.0595 | -0.0809 | -0.0962 | -0.0612 | -0.0069 | -0.0424 | -0.0268 |
| Fast-DetectGPT | -0.1923 | -0.0432 | -0.2700 | -0.2084 | 0.0175 | 0.2316 | -0.0666 |
| Zero-shot Avg. | -0.1744 | 0.0197 | 0.0320 | -0.1969 | 0.0120 | 0.0152 | -0.0332 |
| *Supervised Methods* | | | | | | | |
| GLTR-based | -0.1138 | -0.0848 | -0.2012 | -0.1565 | -0.0224 | -0.0143 | -0.0918 |
| PPL-based | -0.1344 | -0.1186 | -0.1676 | -0.1444 | -0.0093 | -0.0286 | -0.0467 |
| Ghostbuster | -0.0689 | -0.0156 | -0.2041 | -0.0596 | 0.0061 | -0.0736 | 0.0305 |
| Likelihood-based | 0.0736 | -0.0212 | 0.0126 | -0.0735 | 0.0121 | 0.0064 | -0.0014 |
| Rank-based | 0.0296 | 0.0057 | -0.0273 | 0.0445 | 0.0103 | 0.0219 | -0.0295 |
| Logrank-based | 0.0515 | -0.0330 | -0.0080 | 0.0085 | -0.0030 | -0.0016 | -0.0137 |
| Entropy-based | -0.0210 | -0.0176 | 0.0442 | -0.0187 | 0.0089 | -0.0103 | -0.0197 |
| XLNet | -0.1230 | -0.0002 | 0.0186 | 0.0019 | 0.0094 | -0.0318 | -0.0199 |
| DeBERTa | -0.0352 | -0.0334 | -0.0184 | -0.0116 | -0.0007 | -0.0026 | -0.0026 |
| RoBERTa | -0.0037 | -0.0121 | -0.0096 | 0.0002 | -0.0030 | -0.0044 | -0.0054 |
| DistilBERT | -0.0065 | -0.0136 | -0.0077 | -0.0023 | 0.0014 | 0.0020 | -0.0090 |
| XLM-RoBERTa | -0.0017 | -0.0028 | -0.0044 | 0.0013 | -0.0051 | 0.0037 | -0.0071 |
| Mistral | -0.0152 | -0.0043 | -0.0418 | -0.0113 | -0.0036 | 0.0547 | -0.0304 |
| Llama | -0.0057 | -0.0012 | -0.0069 | -0.0010 | -0.0056 | 0.0241 | -0.0412 |
| ImBD | -0.1195 | -0.0773 | -0.0753 | -0.0805 | -0.0137 | 0.0215 | -0.0112 |
| DPIC | -0.0136 | 0.0038 | -0.0112 | 0.0007 | -0.0109 | 0.0229 | 0.0045 |
| DetectAnyLLM | -0.0066 | -0.0001 | -0.0155 | 0.0017 | -0.0073 | 0.0281 | -0.0430 |
| MPU | -0.0100 | -0.1029 | 0.0517 | -0.0135 | 0.0058 | -0.0116 | -0.0069 |
| Supervised Avg. | -0.0686 | -0.0591 | -0.0985 | -0.0822 | -0.0035 | -0.0006 | -0.0191 |
| Overall Avg. | -0.0639 | -0.0433 | -0.1007 | -0.0787 | -0.0027 | 0.0199 | -0.0227 |

Table C.15: The detection performance (AUC) of various detectors in Op-Co experiments.

| AUC | summary | | refine | | expand | |
|---|---|---|---|---|---|---|
| | **polish** | **rewrite** | **polish** | **rewrite** | **polish** | **rewrite** |
| *Zero-shot Methods* | | | | | | |
| Likelihood | 0.0791 | 0.0981 | 0.0473 | 0.0578 | -0.0572 | -0.1036 |
| Rank | 0.0528 | 0.0815 | 0.0686 | 0.0615 | -0.0140 | -0.0295 |
| Logrank | 0.0769 | 0.0934 | 0.0453 | 0.0504 | -0.0684 | -0.1158 |
| Entropy | -0.0634 | -0.0807 | -0.0334 | -0.0422 | 0.0459 | 0.0812 |
| DetectGPT | 0.1141 | 0.1507 | 0.0739 | 0.0438 | 0.0198 | -0.0128 |
| Fast-DetectGPT | 0.1092 | 0.1308 | 0.0906 | 0.1046 | -0.0930 | -0.1186 |
| Zero-shot Avg. | 0.0615 | 0.0790 | 0.0487 | 0.0460 | -0.0278 | -0.0499 |
| *Supervised Methods* | | | | | | |
| GLTR-based | -0.0642 | -0.0665 | -0.0435 | -0.0295 | -0.0365 | -0.0502 |
| PPL-based | -0.0598 | -0.0868 | 0.0115 | 0.0130 | -0.0626 | -0.0478 |
| Ghostbuster | -0.1239 | -0.0701 | -0.0660 | -0.0954 | -0.0374 | -0.0475 |
| Likelihood-based | -0.1639 | -0.1936 | -0.2430 | -0.2182 | 0.0265 | -0.0282 |
| Rank-based | -0.2146 | -0.2166 | -0.3111 | -0.3340 | -0.0050 | -0.0335 |
| Logrank-based | -0.2604 | -0.1980 | -0.3346 | -0.3217 | -0.0122 | -0.0296 |
| Entropy-based | -0.0926 | -0.0216 | -0.1765 | -0.1384 | -0.0063 | 0.0523 |
| XLNet | -0.0005 | -0.0037 | -0.0090 | -0.0017 | 0.0011 | 0.0026 |
| DeBERTa | -0.0030 | -0.0005 | -0.0049 | 0.0001 | 0.0012 | 0.0004 |
| RoBERTa | -0.0037 | -0.0121 | -0.0096 | 0.0002 | -0.0030 | -0.0044 |
| DistilBERT | -0.0065 | -0.0136 | -0.0077 | -0.0023 | 0.0014 | 0.0020 |
| XLM-RoBERTa | -0.0017 | -0.0028 | -0.0044 | 0.0013 | -0.0051 | 0.0037 |
| Mistral | -0.0152 | -0.0043 | -0.0418 | -0.0113 | -0.0036 | 0.0547 |
| Llama | -0.0057 | -0.0012 | -0.0069 | -0.0010 | -0.0056 | 0.0241 |
| ImBD | -0.1195 | -0.0773 | -0.0753 | -0.0805 | -0.0137 | 0.0215 |
| DPIC | -0.0610 | -0.0784 | -0.0347 | -0.0604 | -0.0112 | -0.0125 |
| DetectAnyLLM | -0.0154 | 0.0071 | 0.0100 | 0.0455 | 0.0027 | -0.0059 |
| MPU | 0.0021 | -0.0151 | -0.0589 | -0.0709 | 0.0225 | 0.0343 |
| Supervised Avg. | -0.0372 | -0.0336 | -0.0112 | -0.0102 | -0.0273 | -0.0835 |
| Overall Avg. | -0.0125 | -0.0054 | 0.0038 | 0.0039 | -0.0275 | -0.0751 |

Table C.16: The detection performance (AUC) of various detectors in Op-Co experiments.

| AUC | translate | | | |
|---|---|---|---|---|
| | **polish** | **expand** | **refine** | **rewrite** |
| *Zero-shot Methods* | | | | |
| Likelihood | 0.0002 | 0.1217 | -0.0665 | 0.0059 |
| Rank | 0.0162 | 0.1696 | -0.0669 | 0.0021 |
| Logrank | 0.0012 | 0.1295 | -0.0733 | -0.0007 |
| Entropy | -0.0124 | -0.0978 | 0.0440 | -0.0208 |
| DetectGPT | 0.0256 | 0.0125 | -0.0487 | 0.0344 |
| Fast-DetectGPT | -0.0337 | 0.1932 | -0.1523 | -0.0411 |
| Zero-shot Avg. | -0.0005 | 0.0881 | -0.0606 | -0.0034 |
| *Supervised Methods* | | | | |
| GLTR-based | 0.0356 | -0.0022 | -0.0878 | 0.0182 |
| PPL-based | -0.0250 | -0.0322 | -0.0321 | 0.0057 |
| Ghostbuster | 0.0516 | 0.0832 | -0.0093 | 0.0712 |
| Likelihood-based | -0.0383 | -0.1065 | -0.0485 | -0.0052 |
| Rank-based | -0.0327 | -0.0280 | -0.0336 | -0.0108 |
| Logrank-based | -0.0229 | -0.1324 | -0.0320 | -0.0238 |
| Entropy-based | -0.0041 | -0.0231 | -0.0175 | 0.0351 |
| XLNet | -0.0005 | 0.0508 | 0.0103 | 0.0385 |
| DeBERTa | -0.0352 | -0.0334 | -0.0184 | -0.0116 |
| RoBERTa | -0.0046 | -0.0030 | -0.0126 | -0.0011 |
| DistilBERT | -0.0155 | 0.0069 | -0.0182 | 0.0179 |
| XLM-RoBERTa | -0.0261 | -0.0163 | -0.0097 | 0.0088 |
| Mistral | -0.1044 | -0.1381 | -0.0603 | -0.1070 |
| Llama | 0.0192 | 0.0736 | 0.0361 | 0.0342 |
| ImBD | -0.0301 | -0.0427 | -0.0080 | -0.0211 |
| DPIC | -0.0596 | -0.1417 | 0.0354 | -0.0263 |
| DetectAnyLLM | 0.0305 | 0.0895 | -0.0233 | 0.0232 |
| MPU | 0.0541 | -0.0750 | 0.0498 | 0.0368 |
| Supervised Avg. | -0.0116 | -0.0261 | -0.0155 | 0.0046 |
| Overall Avg. | -0.0088 | 0.0024 | -0.0268 | 0.0026 |

Table C.17: The detection performance (AUC) of Direct LM-based detectors in CD experiment under different training conditions.

| Model | xsum | wp | pubmed | squad | review | blog | tweets |
|---|---|---|---|---|---|---|---|
| **epochs=4** | | | | | | | |
| XLNet | 0.9699 | 0.9852 | 0.9290 | 0.9025 | 0.9802 | 0.9862 | 0.9767 |
| DeBERTa | 0.9798 | 0.9941 | 0.9821 | 0.9384 | 0.9883 | 0.9912 | 0.9777 |
| RoBERTa | 0.9825 | 0.7503 | 0.9646 | 0.7650 | 0.6358 | 0.9627 | 0.9933 |
| DistilBERT | 0.9433 | 0.9765 | 0.9380 | 0.8934 | 0.9841 | 0.9827 | 0.9851 |
| XLM-RoBERTa | 0.9577 | 0.9750 | 0.9471 | 0.8580 | 0.9859 | 0.9851 | 0.9923 |
| Mistral | 0.9816 | 0.9668 | 0.9802 | 0.8375 | 0.9249 | 0.9668 | 0.3369 |
| Llama | 0.9549 | 0.9162 | 0.9759 | 0.8867 | 0.9502 | 0.9885 | 0.7182 |
| **epochs=2** | | | | | | | |
| XLNet | 0.9628 | 0.9832 | 0.9462 | 0.9248 | 0.9806 | 0.9851 | 0.9624 |
| DeBERTa | 0.9755 | 0.9934 | 0.9762 | 0.9455 | 0.9908 | 0.9919 | 0.9831 |
| RoBERTa | 0.9821 | 0.9334 | 0.9613 | 0.9078 | 0.6957 | 0.9485 | 0.9939 |
| DistilBERT | 0.9350 | 0.9680 | 0.9272 | 0.8803 | 0.9813 | 0.9835 | 0.9903 |
| XLM-RoBERTa | 0.9587 | 0.9786 | 0.9544 | 0.9250 | 0.9874 | 0.9802 | 0.9879 |
| Mistral | 0.9712 | 0.9136 | 0.9712 | 0.7244 | 0.8514 | 0.9456 | 0.4557 |
| Llama | 0.9459 | 0.8880 | 0.9553 | 0.8829 | 0.9387 | 0.9773 | 0.7275 |
| **loss¡0.001** | | | | | | | |
| XLNet | 0.9705 | 0.9649 | 0.9581 | 0.8908 | 0.9751 | 0.9838 | 0.9791 |
| DeBERTa | 0.9748 | 0.9856 | 0.9847 | 0.9511 | 0.9851 | 0.9917 | 0.9930 |
| RoBERTa | 0.9826 | 0.8947 | 0.9523 | 0.7834 | 0.7264 | 0.9810 | 0.9943 |
| DistilBERT | 0.9385 | 0.9793 | 0.9397 | 0.8936 | 0.9822 | 0.9823 | 0.9867 |
| XLM-RoBERTa | 0.9497 | 0.9633 | 0.9470 | 0.8900 | 0.9831 | 0.9617 | 0.9876 |
| Mistral | 0.9798 | 0.9397 | 0.9773 | 0.7327 | 0.9286 | 0.9528 | 0.3851 |
| Llama | 0.9415 | 0.8465 | 0.9795 | 0.9221 | 0.9571 | 0.9876 | 0.6960 |
| **loss¡0.01** | | | | | | | |
| XLNet | 0.9626 | 0.9819 | 0.9506 | 0.9201 | 0.9853 | 0.9823 | 0.9835 |
| DeBERTa | 0.9802 | 0.9926 | 0.9752 | 0.9617 | 0.9895 | 0.9922 | 0.9854 |
| RoBERTa | 0.9830 | 0.9612 | 0.9609 | 0.9103 | 0.7173 | 0.8327 | 0.9773 |
| DistilBERT | 0.9257 | 0.9835 | 0.9316 | 0.8978 | 0.9843 | 0.9853 | 0.9925 |
| XLM-RoBERTa | 0.9486 | 0.9669 | 0.9541 | 0.9305 | 0.9805 | 0.9844 | 0.9932 |
| Mistral | 0.9789 | 0.9820 | 0.9764 | 0.8078 | 0.9008 | 0.9524 | 0.6230 |
| Llama | 0.9428 | 0.9066 | 0.9789 | 0.9230 | 0.9530 | 0.9880 | 0.8016 |

Table C.18: The detection performance (AUC) of Direct LM-based detectors in CM experiment under different training conditions.

| Model | Llama3.1-8b | DeepSeek-chat | gpt4o-mini | Qwen |
|---|---|---|---|---|
| **epochs=4** | | | | |
| XLNet | 0.9466 | 0.9952 | 0.9931 | 0.9843 |
| DeBERTa | 0.9482 | 0.9983 | 0.9980 | 0.9881 |
| RoBERTa | 0.9317 | 0.9968 | 0.9971 | 0.9889 |
| DistilBERT | 0.9508 | 0.9893 | 0.9865 | 0.9753 |
| XLM-RoBERTa | 0.9504 | 0.9945 | 0.9917 | 0.9685 |
| Mistral | 0.9596 | 0.9764 | 0.9855 | 0.9685 |
| Llama | 0.9383 | 0.9854 | 0.9831 | 0.9692 |
| **epochs=2** | | | | |
| XLNet | 0.9405 | 0.9942 | 0.9946 | 0.9818 |
| DeBERTa | 0.9526 | 0.9985 | 0.9979 | 0.9879 |
| RoBERTa | 0.9117 | 0.9979 | 0.9972 | 0.9859 |
| DistilBERT | 0.9360 | 0.9872 | 0.9824 | 0.9741 |
| XLM-RoBERTa | 0.9474 | 0.9962 | 0.9939 | 0.9839 |
| Mistral | 0.9546 | 0.9732 | 0.9770 | 0.9577 |
| Llama | 0.9405 | 0.9782 | 0.9752 | 0.9574 |
| **loss¡0.001** | | | | |
| XLNet | 0.9316 | 0.9962 | 0.9927 | 0.9846 |
| DeBERTa | 0.9293 | 0.9985 | 0.9967 | 0.9849 |
| RoBERTa | 0.9547 | 0.9966 | 0.9960 | 0.9907 |
| DistilBERT | 0.9510 | 0.9864 | 0.9853 | 0.9719 |
| XLM-RoBERTa | 0.9552 | 0.9954 | 0.9838 | 0.9833 |
| Mistral | 0.9528 | 0.9125 | 0.9837 | 0.8994 |
| Llama | 0.9516 | 0.9872 | 0.9833 | 0.9703 |
| **loss¡0.01** | | | | |
| XLNet | 0.9453 | 0.9961 | 0.9952 | 0.9850 |
| DeBERTa | 0.9283 | 0.9984 | 0.9969 | 0.9879 |
| RoBERTa | 0.9071 | 0.9955 | 0.9956 | 0.9862 |
| DistilBERT | 0.9478 | 0.9868 | 0.9872 | 0.9754 |
| XLM-RoBERTa | 0.9464 | 0.9938 | 0.9930 | 0.9847 |
| Mistral | 0.9485 | 0.9771 | 0.9872 | 0.9020 |
| Llama | 0.9524 | 0.9877 | 0.9838 | 0.9706 |

Table C.19: The detection performance (AUC) of Direct LM-based detectors in CO experiment under different training conditions.

| Model | create | rewrite | summary | polish | refine | expand | translate |
|---|---|---|---|---|---|---|---|
| **epochs=4** | | | | | | | |
| XLNet | 0.9953 | 0.9942 | 0.9931 | 0.9941 | 0.9910 | 0.9871 | 0.8548 |
| DeBERTa | 0.9992 | 0.9995 | 0.9968 | 0.9976 | 0.9972 | 0.9969 | 0.9316 |
| RoBERTa | 0.9982 | 0.9981 | 0.9960 | 0.9988 | 0.9984 | 0.9952 | 0.9197 |
| DistilBERT | 0.9968 | 0.9952 | 0.9846 | 0.9807 | 0.9850 | 0.9799 | 0.8363 |
| XLM-RoBERTa | 0.9975 | 0.9964 | 0.9926 | 0.9919 | 0.9943 | 0.9928 | 0.8629 |
| Mistral | 0.9236 | 0.9196 | 0.9154 | 0.9691 | 0.9835 | 0.9853 | 0.8699 |
| Llama | 0.9957 | 0.9928 | 0.9922 | 0.9840 | 0.9844 | 0.9913 | 0.8407 |
| **epochs=2** | | | | | | | |
| XLNet | 0.9976 | 0.9978 | 0.9913 | 0.9921 | 0.9874 | 0.9902 | 0.8837 |
| DeBERTa | 0.9995 | 0.9994 | 0.9976 | 0.9980 | 0.9980 | 0.9977 | 0.9383 |
| RoBERTa | 0.9982 | 0.9994 | 0.9968 | 0.9989 | 0.9981 | 0.9991 | 0.9323 |
| DistilBERT | 0.9970 | 0.9955 | 0.9856 | 0.9800 | 0.9851 | 0.9812 | 0.8509 |
| XLM-RoBERTa | 0.9992 | 0.9983 | 0.9924 | 0.9908 | 0.9936 | 0.9896 | 0.8751 |
| Mistral | 0.9963 | 0.9864 | 0.9860 | 0.9619 | 0.9754 | 0.9816 | 0.8731 |
| Llama | 0.9970 | 0.9886 | 0.9845 | 0.9737 | 0.9675 | 0.9829 | 0.8464 |
| **loss¡0.001** | | | | | | | |
| XLNet | 0.9980 | 0.9952 | 0.9882 | 0.9926 | 0.9875 | 0.9823 | 0.9036 |
| DeBERTa | 0.9989 | 0.9990 | 0.9961 | 0.9971 | 0.9966 | 0.9956 | 0.9482 |
| RoBERTa | 0.9948 | 0.9991 | 0.9957 | 0.9969 | 0.9982 | 0.9943 | 0.8569 |
| DistilBERT | 0.9961 | 0.9956 | 0.9843 | 0.9822 | 0.9847 | 0.9813 | 0.8433 |
| XLM-RoBERTa | 0.9895 | 0.9938 | 0.9946 | 0.9897 | 0.9887 | 0.9887 | 0.8738 |
| Mistral | 0.9231 | 0.9209 | 0.9166 | 0.9125 | 0.9735 | 0.9172 | 0.8493 |
| Llama | 0.9964 | 0.9943 | 0.9912 | 0.9890 | 0.9876 | 0.9883 | 0.8374 |
| **loss¡0.01** | | | | | | | |
| XLNet | 0.9936 | 0.9976 | 0.9915 | 0.9940 | 0.9941 | 0.9816 | 0.9362 |
| DeBERTa | 0.9993 | 0.9993 | 0.9975 | 0.9968 | 0.9963 | 0.9952 | 0.9498 |
| RoBERTa | 0.9963 | 0.9988 | 0.9971 | 0.9970 | 0.9979 | 0.9866 | 0.9443 |
| DistilBERT | 0.9964 | 0.9967 | 0.9926 | 0.9823 | 0.9871 | 0.9819 | 0.8634 |
| XLM-RoBERTa | 0.9990 | 0.9971 | 0.9936 | 0.9932 | 0.9858 | 0.9903 | 0.9065 |
| Mistral | 0.9195 | 0.9194 | 0.9175 | 0.9798 | 0.9720 | 0.9814 | 0.8457 |
| Llama | 0.9942 | 0.9941 | 0.9913 | 0.9801 | 0.9897 | 0.9876 | 0.8384 |

Table C.20: The performace (AUC) of detectors on different training scales in CD experiment, grouped by formality levels (Formal vs. Informal) with overall average.

| scale | model | xsum | pubmed | squad | wp | review | blog | tweets |
|---|---|---|---|---|---|---|---|---|
| 500 | GLTR-based | 0.5262 | 0.5696 | 0.5309 | 0.6504 | 0.5982 | 0.6452 | 0.6153 |
| | PPL-based | 0.5379 | 0.5546 | 0.5841 | 0.6396 | 0.6401 | 0.3906 | 0.7912 |
| | Ghostbuster | 0.5570 | 0.5496 | 0.6201 | 0.6940 | 0.6853 | 0.6906 | 0.6969 |
| | Likelihood-based | 0.4708 | 0.4836 | 0.5167 | 0.6185 | 0.5698 | 0.5257 | 0.6189 |
| | Rank-based | 0.5066 | 0.5126 | 0.5005 | 0.6216 | 0.6002 | 0.6085 | 0.8531 |
| | Logrank-based | 0.4608 | 0.5335 | 0.5120 | 0.6135 | 0.5985 | 0.4983 | 0.8109 |
| | Entropy-based | 0.4466 | 0.5072 | 0.5140 | 0.5341 | 0.5544 | 0.4693 | 0.8964 |
| | XLNet | 0.9380 | 0.8873 | 0.8666 | 0.9413 | 0.9703 | 0.9478 | 0.9664 |
| | DeBERTa | 0.9449 | 0.9459 | 0.9195 | 0.9852 | 0.9711 | 0.9810 | 0.9820 |
| | RoBERTa | 0.9584 | 0.9355 | 0.8704 | 0.9632 | 0.8897 | 0.9477 | 0.9556 |
| | DistilBERT | 0.9356 | 0.9036 | 0.8185 | 0.9524 | 0.9748 | 0.9523 | 0.9802 |
| | XLM-RoBERTa | 0.9223 | 0.9316 | 0.8327 | 0.9486 | 0.9773 | 0.9558 | 0.9742 |
| | Mistral | 0.8684 | 0.8985 | 0.7716 | 0.7920 | 0.7455 | 0.8026 | 0.3728 |
| | Llama | 0.9244 | 0.8820 | 0.8278 | 0.5935 | 0.8667 | 0.8987 | 0.7487 |
| | ImBD | 0.7671 | 0.7699 | 0.7219 | 0.9373 | 0.7979 | 0.9326 | 0.7529 |
| | DPIC | 0.8517 | 0.7806 | 0.7453 | 0.7626 | 0.8001 | 0.8946 | 0.8027 |
| | DetectAnyLLM | 0.8260 | 0.8276 | 0.6798 | 0.7878 | 0.8758 | 0.9409 | 0.9077 |
| | MPU | 0.6555 | 0.4393 | 0.4026 | 0.5003 | 0.4980 | 0.3579 | 0.3952 |
| 1000 | GLTR-based | 0.5365 | 0.5471 | 0.5640 | 0.6325 | 0.6321 | 0.6434 | 0.5578 |
| | PPL-based | 0.5356 | 0.5744 | 0.6232 | 0.6524 | 0.6705 | 0.5719 | 0.8210 |
| | Ghostbuster | 0.5646 | 0.5199 | 0.5949 | 0.6801 | 0.7157 | 0.6855 | 0.5715 |
| | Likelihood-based | 0.4534 | 0.4979 | 0.5325 | 0.6386 | 0.5684 | 0.5028 | 0.6208 |
| | Rank-based | 0.5160 | 0.5326 | 0.5075 | 0.6047 | 0.6061 | 0.5282 | 0.8832 |
| | Logrank-based | 0.4801 | 0.5414 | 0.5281 | 0.6434 | 0.6061 | 0.4942 | 0.8605 |
| | Entropy-based | 0.4549 | 0.5228 | 0.5178 | 0.5524 | 0.5589 | 0.5069 | 0.8726 |
| | XLNet | 0.9410 | 0.9083 | 0.9052 | 0.9681 | 0.9741 | 0.9801 | 0.9123 |
| | DeBERTa | 0.9573 | 0.9654 | 0.9403 | 0.9909 | 0.9790 | 0.9896 | 0.9899 |
| | RoBERTa | 0.9801 | 0.9285 | 0.8596 | 0.9556 | 0.8651 | 0.9775 | 0.9137 |
| | DistilBERT | 0.9438 | 0.9060 | 0.8218 | 0.9643 | 0.9775 | 0.9704 | 0.9880 |
| | XLM-RoBERTa | 0.9317 | 0.9482 | 0.8576 | 0.9687 | 0.9746 | 0.9687 | 0.9830 |
| | Mistral | 0.9020 | 0.9409 | 0.8594 | 0.9136 | 0.8044 | 0.8769 | 0.3198 |
| | Llama | 0.9156 | 0.9371 | 0.8538 | 0.7771 | 0.8993 | 0.9431 | 0.8971 |
| | ImBD | 0.8387 | 0.7688 | 0.7401 | 0.9453 | 0.8353 | 0.9096 | 0.8381 |
| | DPIC | 0.8581 | 0.7983 | 0.7916 | 0.8405 | 0.8495 | 0.8547 | 0.8111 |
| | DetectAnyLLM | 0.8510 | 0.8807 | 0.5185 | 0.7208 | 0.6859 | 0.9603 | 0.9000 |
| | MPU | 0.5592 | 0.3837 | 0.4904 | 0.6592 | 0.4870 | 0.7425 | 0.1083 |

Table C.21: The performace (AUC) of detectors on different training scales in CD experiment, grouped by formality levels (Formal vs. Informal) with overall average.

| scale | model | xsum | pubmed | squad | wp | review | blog | tweets |
|---|---|---|---|---|---|---|---|---|
| | GLTR-based | 0.5292 | 0.5690 | 0.5860 | 0.6699 | 0.6620 | 0.6657 | 0.6952 |
| | PPL-based | 0.5585 | 0.6113 | 0.6516 | 0.6264 | 0.7024 | 0.5926 | 0.8517 |
| | Ghostbuster | 0.5879 | 0.5550 | 0.6086 | 0.5755 | 0.6975 | 0.7239 | 0.6143 |
| | Likelihood-based | 0.4856 | 0.4927 | 0.5369 | 0.6539 | 0.5949 | 0.4812 | 0.6209 |
| | Rank-based | 0.5181 | 0.5375 | 0.4929 | 0.6283 | 0.6074 | 0.5470 | 0.6842 |
| | Logrank-based | 0.5024 | 0.5551 | 0.5404 | 0.6398 | 0.5949 | 0.4614 | 0.7745 |
| | Entropy-based | 0.4671 | 0.5391 | 0.5143 | 0.5498 | 0.5498 | 0.4851 | 0.5865 |
| | XLNet | 0.9568 | 0.9327 | 0.8951 | 0.9761 | 0.9779 | 0.9862 | 0.9461 |
| | DeBERTa | 0.9681 | 0.9730 | 0.9500 | 0.9941 | 0.9861 | 0.9895 | 0.9792 |
| 2000 | RoBERTa | 0.9773 | 0.9566 | 0.8132 | 0.9757 | 0.6401 | 0.9479 | 0.9925 |
| | DistilBERT | 0.9433 | 0.9226 | 0.8751 | 0.9758 | 0.9795 | 0.9799 | 0.9881 |
| | XLM-RoBERTa | 0.9447 | 0.9399 | 0.8913 | 0.9703 | 0.9825 | 0.9745 | 0.9901 |
| | Mistral | 0.9609 | 0.9772 | 0.8320 | 0.9208 | 0.9300 | 0.9372 | 0.5792 |
| | Llama | 0.9291 | 0.9606 | 0.8929 | 0.8813 | 0.9364 | 0.9744 | 0.7610 |
| | ImBD | 0.9027 | 0.7694 | 0.7830 | 0.9300 | 0.8598 | 0.9412 | 0.8825 |
| | DPIC | 0.8761 | 0.8246 | 0.8177 | 0.8696 | 0.8780 | 0.8836 | 0.8614 |
| | DetectAnyLLM | 0.9232 | 0.8941 | 0.5505 | 0.5935 | 0.6645 | 0.9655 | 0.9305 |
| | MPU | 0.7321 | 0.6819 | 0.4612 | 0.6329 | 0.5825 | 0.4424 | 0.5938 |

Table C.22: The performace (AUC) of detectors on different training scales in CM experiment.

| scale | model | Llama3.1-8b | DeepSeek-chat | gpt4o-mini | Qwen-plus |
|---|---|---|---|---|---|
| | GLTR-based | 0.6407 | 0.6894 | 0.6875 | 0.7058 |
| | PPL-based | 0.6424 | 0.7243 | 0.7269 | 0.7222 |
| | Ghostbuster | 0.6978 | 0.7765 | 0.7357 | 0.8043 |
| | Likelihood-based | 0.6281 | 0.6226 | 0.6373 | 0.6161 |
| | Rank-based | 0.6455 | 0.6331 | 0.6451 | 0.6336 |
| | Logrank-based | 0.7215 | 0.6761 | 0.6826 | 0.6882 |
| | Entropy-based | 0.5973 | 0.6044 | 0.6005 | 0.5962 |
| | XLNet | 0.9105 | 0.9712 | 0.9711 | 0.9663 |
| | DeBERTa | 0.9065 | 0.9919 | 0.9895 | 0.9750 |
| 500 | RoBERTa | 0.8366 | 0.9941 | 0.9932 | 0.9812 |
| | DistilBERT | 0.9038 | 0.9639 | 0.9579 | 0.9597 |
| | XLM-RoBERTa | 0.9062 | 0.9816 | 0.9805 | 0.9753 |
| | Mistral | 0.8413 | 0.8808 | 0.8725 | 0.8840 |
| | Llama | 0.8631 | 0.8630 | 0.8892 | 0.8809 |
| | ImBD | 0.6991 | 0.8957 | 0.9069 | 0.8701 |
| | DPIC | 0.8192 | 0.8776 | 0.8407 | 0.8810 |
| | DetectAnyLLM | 0.8722 | 0.8792 | 0.9026 | 0.9152 |
| | MPU | 0.6486 | 0.5352 | 0.5812 | 0.4960 |
| | GLTR-based | 0.6607 | 0.7113 | 0.7121 | 0.7113 |
| | PPL-based | 0.6969 | 0.7515 | 0.7493 | 0.7472 |
| | Ghostbuster | 0.6915 | 0.8171 | 0.7449 | 0.7943 |
| | Likelihood-based | 0.6056 | 0.6302 | 0.6470 | 0.6618 |
| | Rank-based | 0.6394 | 0.6376 | 0.6575 | 0.6509 |
| | Logrank-based | 0.7224 | 0.7028 | 0.6983 | 0.7108 |
| | Entropy-based | 0.5942 | 0.6113 | 0.6049 | 0.6226 |
| | XLNet | 0.8973 | 0.9863 | 0.9777 | 0.9726 |
| | DeBERTa | 0.9372 | 0.9969 | 0.9962 | 0.9866 |
| 1000 | RoBERTa | 0.9008 | 0.9960 | 0.9958 | 0.9854 |
| | DistilBERT | 0.9360 | 0.9788 | 0.9711 | 0.9679 |
| | XLM-RoBERTa | 0.9153 | 0.9906 | 0.9883 | 0.9792 |
| | Mistral | 0.9017 | 0.9317 | 0.9359 | 0.9303 |
| | Llama | 0.9210 | 0.9352 | 0.9307 | 0.9270 |
| | ImBD | 0.7324 | 0.9224 | 0.9203 | 0.8671 |
| | DPIC | 0.8722 | 0.9092 | 0.8668 | 0.8918 |
| | DetectAnyLLM | 0.9201 | 0.9658 | 0.9612 | 0.9584 |
| | MPU | 0.5104 | 0.6379 | 0.5449 | 0.4728 |

Table C.23: The performace (AUC) of detectors on different training scales in CM experiment.

| scale | model | Llama3.1-8b | DeepSeek-chat | gpt4o-mini | Qwen-plus |
|---|---|---|---|---|---|
| | GLTR-based | 0.6777 | 0.7166 | 0.7309 | 0.7339 |
| | PPL-based | 0.7258 | 0.7658 | 0.7728 | 0.7646 |
| | Ghostbuster | 0.7119 | 0.8226 | 0.7643 | 0.7973 |
| | Likelihood-based | 0.6244 | 0.6407 | 0.6675 | 0.6692 |
| | Rank-based | 0.6692 | 0.6542 | 0.6709 | 0.6690 |
| | Logrank-based | 0.7426 | 0.7051 | 0.7121 | 0.7229 |
| | Entropy-based | 0.5976 | 0.6279 | 0.6103 | 0.6247 |
| | XLNet | 0.9205 | 0.9939 | 0.9887 | 0.9793 |
| 2000 | DeBERTa | 0.9234 | 0.9981 | 0.9970 | 0.9867 |
| | RoBERTa | 0.9081 | 0.9967 | 0.9970 | 0.9858 |
| | DistilBERT | 0.9394 | 0.9818 | 0.9814 | 0.9722 |
| | XLM-RoBERTa | 0.9348 | 0.9947 | 0.9893 | 0.9797 |
| | Mistral | 0.9543 | 0.9775 | 0.9850 | 0.9696 |
| | Llama | 0.9360 | 0.9716 | 0.9699 | 0.9628 |
| | ImBD | 0.7487 | 0.9394 | 0.9488 | 0.9011 |
| | DPIC | 0.8826 | 0.9338 | 0.8911 | 0.9151 |
| | DetectAnyLLM | 0.9129 | 0.9744 | 0.9739 | 0.9647 |
| | MPU | 0.5093 | 0.5652 | 0.5032 | 0.4552 |

Table C.24: The performace (AUC) of detectors on different training scales in CO experiment, grouped by operation types.

| scale | model | create | rewrite | summary | polish | refine | expand | translate |
|---|---|---|---|---|---|---|---|---|
| | GLTR-based | 0.7180 | 0.6626 | 0.6692 | 0.6583 | 0.5826 | 0.7291 | 0.6292 |
| | PPL-based | 0.7165 | 0.6723 | 0.7112 | 0.6964 | 0.6879 | 0.7274 | 0.6611 |
| | Ghostbuster | 0.7362 | 0.7257 | 0.5765 | 0.6893 | 0.7265 | 0.6592 | 0.6609 |
| | Likelihood-based | 0.5320 | 0.6172 | 0.6288 | 0.6091 | 0.5843 | 0.5674 | 0.5826 |
| | Rank-based | 0.4669 | 0.6147 | 0.6306 | 0.6057 | 0.5937 | 0.5947 | 0.6462 |
| | Logrank-based | 0.4804 | 0.6616 | 0.7564 | 0.6772 | 0.6787 | 0.5928 | 0.6896 |
| | Entropy-based | 0.5230 | 0.5934 | 0.5786 | 0.5957 | 0.5594 | 0.6103 | 0.5634 |
| | XLNet | 0.9937 | 0.9754 | 0.9677 | 0.9520 | 0.9611 | 0.9611 | 0.7907 |
| 500 | DeBERTa | 0.9990 | 0.9967 | 0.9889 | 0.9913 | 0.9838 | 0.9875 | 0.8460 |
| | RoBERTa | 0.9964 | 0.9964 | 0.9825 | 0.9892 | 0.9901 | 0.9703 | 0.8968 |
| | DistilBERT | 0.9921 | 0.9784 | 0.9595 | 0.9420 | 0.9621 | 0.9518 | 0.7889 |
| | XLM-RoBERTa | 0.9961 | 0.9905 | 0.9718 | 0.9771 | 0.9725 | 0.9770 | 0.8518 |
| | Mistral | 0.9900 | 0.8787 | 0.9274 | 0.8160 | 0.8378 | 0.9337 | 0.7251 |
| | Llama | 0.9798 | 0.9346 | 0.8835 | 0.8855 | 0.8668 | 0.9571 | 0.7525 |
| | ImBD | 0.9854 | 0.8485 | 0.8482 | 0.8316 | 0.7938 | 0.8203 | 0.7380 |
| | DPIC | 0.8152 | 0.8634 | 0.8572 | 0.8118 | 0.8364 | 0.8632 | 0.7253 |
| | DetectAnyLLM | 0.9551 | 0.9618 | 0.8105 | 0.9101 | 0.8784 | 0.9261 | 0.7989 |
| | MPU | 0.7520 | 0.6529 | 0.6699 | 0.6009 | 0.4493 | 0.5467 | 0.5329 |
| | GLTR-based | 0.7139 | 0.6673 | 0.6412 | 0.6663 | 0.5785 | 0.7254 | 0.6363 |
| | PPL-based | 0.7358 | 0.7143 | 0.7561 | 0.7042 | 0.7113 | 0.7460 | 0.6798 |
| | Ghostbuster | 0.7340 | 0.7373 | 0.7774 | 0.6885 | 0.7561 | 0.6639 | 0.7029 |
| | Likelihood-based | 0.5234 | 0.6114 | 0.6437 | 0.6190 | 0.6106 | 0.5874 | 0.6079 |
| | Rank-based | 0.5137 | 0.6042 | 0.6601 | 0.6165 | 0.6162 | 0.6087 | 0.6373 |
| | Logrank-based | 0.4841 | 0.6693 | 0.7751 | 0.6985 | 0.7019 | 0.6041 | 0.7022 |
| | Entropy-based | 0.5788 | 0.5858 | 0.5514 | 0.6011 | 0.5669 | 0.5970 | 0.5681 |
| | XLNet | 0.9962 | 0.9962 | 0.9842 | 0.9821 | 0.9790 | 0.9774 | 0.8641 |
| 1000 | DeBERTa | 0.9994 | 0.9988 | 0.9924 | 0.9950 | 0.9945 | 0.9923 | 0.8655 |
| | RoBERTa | 0.9971 | 0.9977 | 0.9908 | 0.9964 | 0.9958 | 0.9950 | 0.9072 |
| | DistilBERT | 0.9943 | 0.9900 | 0.9675 | 0.9685 | 0.9732 | 0.9664 | 0.8317 |
| | XLM-RoBERTa | 0.9948 | 0.9903 | 0.9849 | 0.9844 | 0.9842 | 0.9835 | 0.8787 |
| | Mistral | 0.9944 | 0.9580 | 0.9755 | 0.9060 | 0.9152 | 0.9537 | 0.7981 |
| | Llama | 0.9934 | 0.9627 | 0.9454 | 0.9317 | 0.9437 | 0.9601 | 0.8133 |
| | ImBD | 0.9922 | 0.8980 | 0.8892 | 0.8845 | 0.8458 | 0.8390 | 0.7601 |
| | DPIC | 0.8445 | 0.9023 | 0.8675 | 0.8437 | 0.8369 | 0.8677 | 0.7155 |
| | DetectAnyLLM | 0.9860 | 0.9775 | 0.9322 | 0.9763 | 0.9619 | 0.9800 | 0.8564 |
| | MPU | 0.5272 | 0.5014 | 0.5184 | 0.5442 | 0.4189 | 0.5244 | 0.4242 |

Table C.25: The performace (AUC) of detectors on different training scales in CO experiment, grouped by operation types.

| scale | model | create | rewrite | summary | polish | refine | expand | translate |
|---|---|---|---|---|---|---|---|---|
| | GLTR-based | 0.6811 | 0.6793 | 0.6762 | 0.6739 | 0.5894 | 0.7305 | 0.6485 |
| | PPL-based | 0.77 | 0.7388 | 0.7656 | 0.7418 | 0.7189 | 0.7719 | 0.6903 |
| | Ghostbuster | 0.7496 | 0.7682 | 0.6387 | 0.7037 | 0.7156 | 0.6980 | 0.7015 |
| | Likelihood-based | 0.5644 | 0.6342 | 0.6636 | 0.6105 | 0.6258 | 0.5987 | 0.6209 |
| | Rank-based | 0.6012 | 0.6358 | 0.6827 | 0.6226 | 0.6395 | 0.6136 | 0.6466 |
| | Logrank-based | 0.5395 | 0.7010 | 0.7786 | 0.6906 | 0.7229 | 0.6129 | 0.7184 |
| | Entropy-based | 0.5610 | 0.5995 | 0.5835 | 0.6068 | 0.5845 | 0.5785 | 0.5838 |
| | XLNet | 0.9984 | 0.9942 | 0.9898 | 0.9904 | 0.9873 | 0.9881 | 0.8573 |
| 2000 | DeBERTa | 0.9994 | 0.9988 | 0.9931 | 0.9966 | 0.9961 | 0.9962 | 0.9200 |
| | RoBERTa | 0.9973 | 0.9980 | 0.9938 | 0.9968 | 0.9969 | 0.9937 | 0.9460 |
| | DistilBERT | 0.9950 | 0.9934 | 0.9775 | 0.9739 | 0.9831 | 0.9762 | 0.8368 |
| | XLM-RoBERTa | 0.9941 | 0.9870 | 0.9896 | 0.9884 | 0.9916 | 0.9903 | 0.9172 |
| | Mistral | 0.9951 | 0.9879 | 0.9895 | 0.9590 | 0.9829 | 0.9799 | 0.8742 |
| | Llama | 0.9963 | 0.9910 | 0.9857 | 0.9772 | 0.9812 | 0.9813 | 0.8386 |
| | ImBD | 0.9928 | 0.9537 | 0.9199 | 0.9354 | 0.8679 | 0.8821 | 0.7327 |
| | DPIC | 0.8942 | 0.9282 | 0.9095 | 0.8790 | 0.9055 | 0.9143 | 0.7704 |
| | DetectAnyLLM | 0.9835 | 0.9813 | 0.9542 | 0.9811 | 0.9689 | 0.9828 | 0.8756 |
| | MPU | 0.5007 | 0.5258 | 0.4320 | 0.5936 | 0.4817 | 0.6509 | 0.6013 |

Table C.26: The detection performance (AUC) of open-access detectors in CD experiment.

| AUC | Formal | | | | Informal | | | | | Overall Avg. |
|---|---|---|---|---|---|---|---|---|---|---|
| | xsum | pubmed | squad | Avg. | wp | review | blog | tweets | Avg. | |
| ArguGPT | 0.6059 | 0.7674 | 0.7263 | 0.6999 | 0.8496 | 0.8724 | 0.9106 | 0.8610 | 0.8734 | 0.7990 |
| RADAR | 0.8703 | 0.7628 | 0.6954 | 0.7762 | 0.5438 | 0.6663 | 0.7051 | 0.6615 | 0.6776 | 0.7007 |
| OpenAI | 0.5327 | 0.5178 | 0.4870 | 0.5125 | 0.5505 | 0.4575 | 0.5725 | 0.5658 | 0.5319 | 0.5263 |
| MPU | 0.6845 | 0.6844 | 0.6359 | 0.6683 | 0.7530 | 0.5666 | 0.9037 | 0.8260 | 0.7654 | 0.7220 |
| RoBerta-HC3 | 0.3571 | 0.3673 | 0.4482 | 0.3909 | 0.8994 | 0.4899 | 0.9615 | 0.9711 | 0.8075 | 0.6421 |

Table C.27: The detection performance (AUC) of open-access detectors in CM experiment.

| AUC | Llama3.1-8b | Deepseek-chat | gpt4o-mini | Qwen-plus | Avg. |
|---|---|---|---|---|---|
| ArguGPT | 0.6234 | 0.7649 | 0.7962 | 0.7733 | 0.7395 |
| RADAR | 0.6746 | 0.5998 | 0.6532 | 0.6163 | 0.6360 |
| OpenAI | 0.5428 | 0.5140 | 0.5134 | 0.4818 | 0.5130 |
| MPU | 0.5809 | 0.6281 | 0.7407 | 0.6683 | 0.6545 |
| RoBerta-HC3 | 0.4946 | 0.6204 | 0.7177 | 0.6873 | 0.6300 |

Table C.28: The detection performance (AUC) of open-access detectors in CO experiment.

| AUC | Creative Reformation | | | | Linguistic Enhancement | | | | Cross-lingual | Overall Avg. |
|---|---|---|---|---|---|---|---|---|---|---|
| | create | rewrite | summary | Avg. | polish | refine | expand | Avg. | translate | |
| ArguGPT | 0.8959 | 0.7362 | 0.7310 | 0.7877 | 0.7294 | 0.6653 | 0.7695 | 0.7214 | 0.6410 | 0.7383 |
| RADAR | 0.7941 | 0.5939 | 0.7567 | 0.7149 | 0.5993 | 0.5924 | 0.5882 | 0.5933 | 0.5737 | 0.6426 |
| OpenAI | 0.5466 | 0.4541 | 0.6231 | 0.5413 | 0.4854 | 0.4865 | 0.3864 | 0.4528 | 0.6080 | 0.5129 |
| MPU | 0.7571 | 0.6234 | 0.6687 | 0.6831 | 0.6761 | 0.5910 | 0.6360 | 0.6344 | 0.6278 | 0.6543 |
| RoBerta-HC3 | 0.7008 | 0.5700 | 0.6960 | 0.6556 | 0.6391 | 0.5834 | 0.5442 | 0.5889 | 0.6774 | 0.6301 |

