# OpenReview forum: "BRED: A Comprehensive Benchmark for the Robust Evaluation of LLM-Generated Text Detection in Realistic Scenarios"
_ICLR.cc/2026/Conference — ICLR 2026 Conference Withdrawn Submission_

### Official Review · Reviewer_2Pt8 · 2025-10-29

**Soundness:** 3
**Presentation:** 2
**Contribution:** 3
**Rating:** 4
**Confidence:** 4

**Summary:**

This paper introduces a new benchmark for machine-generated text detection , **BRED.** The benchmark aims to systematically, comprehensively, and realistically evaluate the robustness and generalization ability of various LLM-based detectors.

BRED includes **7 domains, 7 text operations**, and **10 LLMs** across four model families (GPT, Qwen, DeepSeek, and LLaMA). It also supports cross-model (LLM-Crossover) and cross-operation (Operation-Crossover) experiments, simulating realistic “multi-stage generation–editing” workflows.

**Strengths:**

- The benchmark covers diverse domains, models, and text operations. On top of this, it designs a comprehensive set of evaluation tasks, including Cross-Domain (CD), Cross-Model (CM), Cross-Operation (CO), Multi-Length (ML), LLM-Variant (LLM-Va), and generation combination tasks (LLM-Co, Op-Co).
- The paper identifies several interesting findings, such as: instruction-tuned and reasoning-mode texts exhibit more “human-like” features and are harder to detect; secondary generation tends to reduce detectability, providing useful insights for training and deploying text detectors in real scenarios.

**Weaknesses:**

- From a dataset perspective, this work indeed contributes new data to the community. However, from the construction viewpoint, it mainly expands the scope of domains, models, and operations. The core contribution lies in the LLM-Co and Op-Co settings, so I suggest the authors focus more on analyzing these parts , e.g., how different operation types, operation counts, and generalization between operations affect detection. Yet, the paper does not provide deeper analysis on these aspects. Meanwhile, the Instruct and Reasoning analyses, which occupy substantial space in the main text, offer relatively limited insights.
- The boundaries among the seven operations (create, rewrite, summary, refine, polish, expand, translate) are not clearly defined. For instance, *rewrite*, *refine*, and *polish* seem rather similar, and *refine* also involves “condensing,” which raises the question of how it is distinct from *summary*.
- Some claims are not rigorously supported. For example, line 354 states that “Reasoning-enhanced texts are harder to detect.” However, as Table 3c shows, the differences between reasoning and non-reasoning texts across GPT, Qwen, and DeepSeek models are only 0.78%, 0.1%, and 0.34%, respectively—less than 1%. Therefore, such a statement seems overstated, especially compared to line 311, where the large–small model AUROC gap is 1.21%, appropriately described as “no significant impact.”
- Typo: line 1036, “large language model (LLM)” should simply be “LLM.”
- Some descriptions lack clarity. For instance, line 190 mentions “12 distinct model combinations,” and line 196 refers to “17 distinct operational combinations”, but what exactly are these? Moreover, the overall dataset distribution, total data size, and train/test splits for each subtask are unclear, making it easy for readers to get lost in the paper.
- Overall, I acknowledge the substantial effort invested in this work. However, the paper requires significant revision to highlight its main contribution while de‑emphasizing analyses that largely replicate prior benchmarks (e.g., ML, CD, CM). This does not mean such analyses should be removed, but rather streamlined (since they do not yield fundamentally new insights) to leave more space for discussing the truly novel aspects, particularly LLM-Co and Op-Co. Supplementary sections could host the repetitive benchmark results.

**Questions:**

- Could the authors clarify the difference between the Likelihood detector in Zero‑shot Methods and the one in Supervised Methods?
- Were all detectors trained and tested on exactly the same data distributions?
- During dataset construction, did the authors adopt any strategies to balance text length distributions across different domains?
- How would additional rounds of model operations (i.e., more than two steps) influence detection performance?

---

### Official Review · Reviewer_PZBQ · 2025-10-30

**Soundness:** 2
**Presentation:** 2
**Contribution:** 2
**Rating:** 2
**Confidence:** 4

**Summary:**

This paper presents BRED, a new benchmark for evaluating machine-generated text detectors in more realistic scenarios. To this aim, the authors leverage 7 different domains (both formal and informal) to produce machine-generated texts using 10 LLMs. The proposed benchmark is hence used to evaluate the performance of zero-shot and supervised detectors. The authors eventually provide some insights into the impact of model size, instruct vs. base models, as well as the impact of reasoning models on detectability.

**Strengths:**

- The proposed benchmark covers a multitude of domains, operations, and model architectures, providing a comprehensive overview of current detectors' capabilities.
- The set of operations is interesting and worth investigating, as it covers the most common operations used for AI-based text editing and similar tasks.
- Having a new benchmark that introduces novel tasks for detecting machine-generated text is valuable given the proliferation of these texts across multiple fields.

**Weaknesses:**

- The proposed suite of benchmark tasks and related findings is not totally novel. For instance, there already exist works [r1] discussing the impact of model size on detectability and secondary generations, but these are not even cited in the manuscript.
- The proposed dataset benchmark lacks adequate summary presentation: what is the final size? What is the distribution of tasks and classes? Overall, the entire presentation is not properly organized and requires proofreading and enhancement in order to fully appreciate the contributions.
- Despite the set of evaluated detectors being reasonable, most state-of-the-art or relevant approaches are missing and not even mentioned. Among these, there are Binoculars [r2], ConDA [r3], WhosAI [r4], and DeTeCtive [r5].
- The cross-model and cross-operation tasks are not properly explained, and some additional details should be provided to the reader. Similarly, what is the (slight) difference between polish and refine in the Cross-Operation tasks? All these details would enhance the readability of the manuscript.
- Most claims are not properly supported: e.g., Llama seems to be harder to detect in Section 5.1, but no reason is provided. Is it due to better overlap with human style or what?
- Some results seem weak to me: for instance, the discrepancies highlighted in Table 3 as tangible for certain cases actually seem not robust to me. The authors should better frame them and properly quantify whether and to what extent these are significant.
- In Section 5.4.1, one of the recommendations suggests using entropy for the Llama family. But this is sort of a circuit, as you should know, text has been machine-generated, and by Llama (i.e., Authorship Attribution task in Machine-generated Text).


[r1] "OpenTuringBench: An Open-Model-based Benchmark and Framework for Machine-Generated Text Detection and Attribution." _arXiv preprint arXiv:2504.11369_ (2025).

[r2] "Spotting LLMs With Binoculars: Zero-Shot Detection of Machine-Generated Text." _International Conference on Machine Learning_. PMLR, 2024.

[r3] "ConDA: Contrastive Domain Adaptation for AI-generated Text Detection." _Proceedings of the 13th International Joint Conference on Natural Language Processing and the 3rd Conference of the Asia-Pacific Chapter of the Association for Computational Linguistics (Volume 1: Long Papers)_. 2023.

[r4] "Is contrasting all you need? contrastive learning for the detection and attribution of ai-generated text." European Conference on Artificial Intelligence (ECAI 2024).

[r5] "Detective: Detecting ai-generated text via multi-level contrastive learning." _Advances in Neural Information Processing Systems_ 37 (2024): 88320-88347.

**Questions:**

Is the final label associated with data binary or multi-class (i.e., suitable for authorship attribution)? If binary, how do the author determine the final label in the presence of LLM-Co tasks?
- Table 2 and the results discussed might be aggregated better. As they are now, it seems that supervised approaches are modest on average, as the authors aggregate linguistic-based and model-based supervised techniques in a single view. It would be better to separate them, given that e.g., RoBERTa-based detectors are very effective.
- Can the authors statistically evaluate the correlation between text length and AUC shown in Figure 2 and discussed in Section 5.1? It does not seem to hold for all approaches.
- Figure 2 (c) requires proofreading: there are some placeholder labels to be fixed, as well as y-ticks are overlapping and unreadable. Also, to which score do these heatmaps (also Figure 2 (b)) refer?
- What are blue/red cells as well as greyed values (see overall avg. for tweets) in Table 2?
- Overall, Figures are difficult to read as excessively small or packed.
- Another key limitations the authors should discuss is multilinguality, as only relying on English might prevent a broader adoption of the proposed benchmark.

---

### Official Review · Reviewer_CZmJ · 2025-11-01

**Soundness:** 2
**Presentation:** 3
**Contribution:** 1
**Rating:** 2
**Confidence:** 5

**Summary:**

The paper proposes a comprehensive benchmark for LLM-generated text detection. The benchmark consists of diverse domains, and compositional operations. The paper also provides extensive experiments on different detectors.

**Strengths:**

AI-generated text detection is an important problem, and creating a comprehensive high benchmark for this problem is of great importance.
The proposed benchmark involves diverse domains, contributing to its realness and comprehensiveness. The paper also conducts extensive experiments.

**Weaknesses:**

Overall, the proposed benchmark and the experimental pipeline lack novelty. Different domains have already been studied and included in existing benchmarks (e.g., [1]). The paper needs to clearly distinguish the proposed benchmark from the existing benchmarks and compare it against them. Also, I highly suggest that the paper be submitted to benchmarking tracks.
Moreover, the paper claims that the proposed benchmark is designed for the robust evaluation of the detectors. However, besides involving different domains, it is unclear how the proposed benchmark contributes to the robustness of the evaluations.
Finally, the paper needs to include other types of detectors (e.g., watermarking [2]) to ensure the generalizability of the results.



[1] Pudasaini, Shushanta, et al. "Benchmarking AI Text Detection: Assessing Detectors Against New Datasets, Evasion Tactics, and Enhanced LLMs." Proceedings of the 1stWorkshop on GenAI Content Detection (GenAIDetect). 2025.

[2] Zhao, Xuandong, et al. "Provable robust watermarking for ai-generated text." ICLR 2024.

**Questions:**

Please refer to the weaknesses.

---

### Official Review · Reviewer_fFrK · 2025-11-01

**Soundness:** 2
**Presentation:** 2
**Contribution:** 1
**Rating:** 4
**Confidence:** 4

**Summary:**

This paper introduces a new and comprehensive benchmark, BRED, aimed at evaluating the detection of text generated by LLMs in realistic scenarios. The design of BRED covers 7 different domains and 7 common text operations, creating a dataset that closely mirrors real-world applications. Using BRED, the paper evaluates texts generated by models of varying scales from four mainstream LLM families. The experiments systematically tested various types of detectors (from zero-shot statistics to open-access detectors) and provides an in-depth analysis. The findings demonstrate the performance, robustness, and limitations of existing detectors across different scenarios.

**Strengths:**

1. Compared with existing benchmarks, BRED offers broader and deeper coverage, incorporating 7 domains and 7 realistic text operations to create a testing environment that closely mirrors real-world scenarios.
2. The benchmark provides a detailed snapshot of the current landscape of AI-generated text detectors, with comprehensive experimental results.
3. The LLM-Crossover and Operation-Crossover designs in this paper, which simulate multi-step and mixed text generation workflows, are particularly innovative in evaluating detector robustness.

**Weaknesses:**

1. Insufficient analytical depth. The analysis could be strengthened with more quantitative evidence supporting the motivation for mixed operations. For example, measuring linguistic properties (e.g., lexical diversity) might explain why certain model or operation combinations significantly affect detectability.
2. Choice of pivot language in back-translation: The “Translation” operation is implemented as English → Chinese → English. Was Chinese selected for specific linguistic reasons, or merely for convenience? Would using other languages (e.g., German) yield notably different results? This hurts the comprehensiveness of the translation operation in data creation.
3. Lack of statistical tests. The conclusion relies on direct AUROC comparisons, claiming that “*large models was slightly higher than for smaller models (AUROC of around 76.07%), the difference was minimal and likely not statistically significant.*”). It would strengthen the rigor of the paper to conduct formal significance tests (e.g., t-test, bootstrap) to confirm or refute such claims, especially for a benchmark paper aiming for compare model capability.
4. Presentation improvements needed:
    - Clarify abbreviations such as “rs” in Table 3 (likely meaning reasoning) directly in the caption.
    - Figure 2(a) combines different detector categories in one plot, making comparison difficult; consider improving readability.
    - Highlight key results in tables (e.g., bold or color) to emphasize main findings.

**Questions:**

All generations are produced with temperature = 0.7, which overlooks real-world decoding diversity (e.g., greedy, top-k, or varying temperatures). Given that decoding strategies strongly influence text properties, do you expect your conclusions to generalize across such variations?

---

### Note · Authors · 2025-12-27

I have read and agree with the venue's withdrawal policy on behalf of myself and my co-authors.